# BAYESIMP: Uncertainty Quantification for Causal Data Fusion

**Siu Lun Chau**[*]
University of Oxford

**Jean-François Ton**[*]
University of Oxford

**Javier González**
Microsoft Research Cambridge

**Yee Whye Teh**
University of Oxford

**Dino Sejdinovic**
University of Oxford

## Abstract

While causal models are becoming one of the mainstays of machine learning, the problem of uncertainty quantification in causal inference remains challenging. In this paper, we study the causal data fusion problem, where datasets pertaining to multiple causal graphs are combined to estimate the average treatment effect of a target variable. As data arises from multiple sources and can vary in quality and quantity, principled uncertainty quantification becomes essential. To that end, we introduce Bayesian Interventional Mean Processes, a framework which combines ideas from probabilistic integration and kernel mean embeddings to represent interventional distributions in the reproducing kernel Hilbert space, while taking into account the uncertainty within each causal graph. To demonstrate the utility of our uncertainty estimation, we apply our method to the Causal Bayesian Optimisation task and show improvements over state-of-the-art methods.

## 1 Introduction

Causal inference has seen a significant surge of research interest in areas such as healthcare [1], ecology [2], and optimisation [3]. However, data fusion, the problem of merging information from multiple data sources, has received limited attention in the context of causal modelling, yet presents significant potential benefits for practical situations [4, 5]. In this work, we consider a causal data fusion problem where two causal graphs are combined for the purposes of inference of a target variable (see Fig.1). In particular, our goal is to quantify the uncertainty under such a setup and determine the level of confidence in our treatment effect estimates.

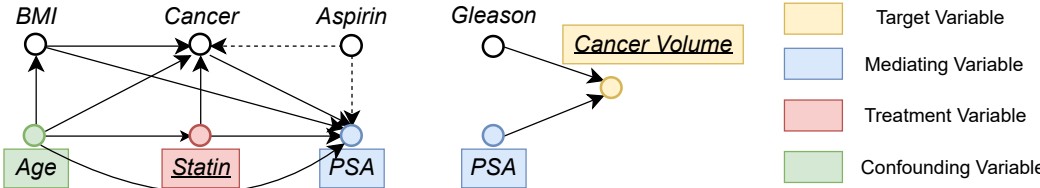

Figure 1: Example problem setup: Causal graphs collected in two separate medical studies i.e. [6] and [7]. (Left) $\mathcal{D}_1$ : Data describing the causal relationships between statin level and Prostate Specific Antigen (PSA). (Right) $\mathcal{D}_2$ : Data from a prostate cancer study for patients about to receive a radical prostatectomy. Goal: **Model** $\mathbb{E}[\textit{Cancer Volume}|\textit{do(Statin)}]$ while also quantifying its uncertainty.

Let us consider the motivating example in Fig.1, where a medical practitioner is investigating how *prostate cancer volume* is affected by a *statin* drug dosage. We consider the case where the doctor

35th Conference on Neural Information Processing Systems (NeurIPS 2021).

---

[*]Denotes equal contribution with alphabetical ordering

only has access to two separate medical studies describing the quantities of interest. On one hand we have observational data, from one medical study $\mathcal{D}_1$ [1], describing the causal relationship between *statin* level and *prostate specific antigen (PSA)*, and on the other hand we have observational data, from a second study $\mathcal{D}_2$ [7], that looked into the link between *PSA* level and *prostate cancer volume*. The goal is to model the **interventional effect** between our target variable (*cancer volume*) and the treatment variable (*statin*). This problem setting is different from the standard observational scenario as it comes with the following challenges:

- **Unmatched data:** Our goal is to estimate $\mathbb{E}[cancer\ volume|do(statin)]$ but the observed *cancer volume* is not paired with *statin* dosage. Instead, they are related via a mediating variable *PSA*.

- **Uncertainty quantification:** The two studies may be of different data quantity/quality. Furthermore, a covariate shift in the mediating variable, i.e. a difference between its distributions in two datasets, may cause inaccurate extrapolation. Hence, we need to account for uncertainty in both datasets.

Formally, let $X$ be the treatment (*Statin*), $Y$ be the mediating variable (*PSA*) and $T$ our target (*cancer volume*), and our aim is to estimate $\mathbb{E}[T|do(X)]$. The problem of unmatched data in a similar context has been previously considered by [5] using a two-staged regression approach ($X \to Y$ and $Y \to T$). However, uncertainty quantification, despite being essential if our estimates of interventional effects will guide decision-making, has not been previously explored. In particular, it is crucial to quantify the uncertainty in both stages as this takes into account the lack of data in specific parts of the space. Given that we are using different datasets for each stage, there are also two sources of epistemic uncertainties (due to lack of data) as well as two sources of aleatoric uncertainties (due to inherent randomness in $Y$ and $T$) [8] . It is thus natural to consider regression models based on Gaussian Processes (GP) [9], as they are able to model both types of uncertainties. However, as GPs, or any other standard regression models, are designed to model conditional expectations only and will fail to capture the underlying distributions of interest (e.g. if there is multimodality in $Y$ as discussed in [10]). This is undesirable since, as we will see, interventional effect estimation requires accurate estimates of distributions. While one could in principle resort to density estimation methods, this becomes challenging since we typically deal with a number of conditional/ interventional densities.

In this paper, we introduce the framework of *Bayesian Interventional Mean Processes* (BAYESIMP) to circumvent the challenges in the causal data fusion setting described above. BAYESIMP considers kernel mean embeddings [11] for representing distributions in a reproducing kernel Hilbert space (RKHS), in which the whole arsenal of kernel methods can be extended to probabilistic inference (e.g. kernel Bayes rule [12], hypothesis testing [13], distribution regression [14]). Specifically, BAYES-IMP uses kernel mean embeddings to represent the interventional distributions and to analytically marginalise out $Y$, hence accounting for aleatoric uncertainties. Further, BAYESIMP uses GPs to estimate the required kernel mean embeddings from data in a Bayesian manner, which allows to quantify the epistemic uncertainties when representing the interventional distributions. To illustrate the quality of our uncertainty estimates, we apply BAYESIMP to Causal Bayesian Optimisation [15], an efficient heuristic to optimise objective functions of the form $x^* = \arg\min_{x \in \mathcal{X}} \mathbb{E}[T|do(X) = x]$. Our contributions are summarised below:

1. We propose a novel *Bayesian Learning of Conditional Mean Embedding* (BAYESCME) that allows us to estimate conditional mean embeddings in a Bayesian framework.

2. Using BAYESCME, we propose a novel *Bayesian Interventional Mean Process* (BAYESIMP) that allows us to model interventional effect across causal graphs without explicit density estimation, while obtaining uncertainty estimates for $\mathbb{E}[T|do(X) = x]$.

3. We apply BAYESIMP to Causal Bayesian Optimisation, a problem introduced in [15] and show significant improvements over existing state-of-the-art methods.

Note that [16] also considered a causal fusion problem but with a different objective. They focused on extrapolating experimental findings across treatment domains, i.e. inferring $\mathbb{E}[Y|do(X)]$ when only data from $p(Y|do(S))$ is observed, where $S$ is some other treatment variable. In contrast, we focus on modelling combined causal graphs, with a strong emphasis on uncertainty quantification. While [17] considered mapping interventional distributions in the RKHS to model quantities such as $\mathbb{E}[T|do(X)]$, they only considered a frequentist approach, which does not account for epistemic uncertainties.

**Notations.** We denote $X, Y, Z$ as random variables taking values in the non-empty sets $\mathcal{X}, \mathcal{Y}$ and $\mathcal{Z}$ respectively. Let $k_x : \mathcal{X} \times \mathcal{X} \to \mathbb{R}$ be positive definite kernels on $X$ with an associated RKHS $\mathcal{H}_{k_x}$. The corresponding canonical feature map $k_x(x', \cdot)$ is denoted as $\phi_x(x')$. Analogously for $Y$ and $Z$.

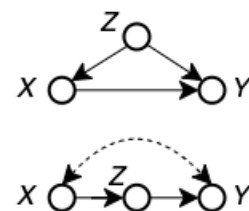

Figure 2: A general two stage causal learning setup.

In the simplest setting, we observe i.i.d samples $\mathcal{D}_1 = \{x_i, y_i, z_i\}_{i=1}^N$ from joint distribution $\mathbb{P}_{XYZ}$ which we concatenate into vectors $\mathbf{x} := [x_1, ..., x_N]^\top$. Similarly for $\mathbf{y}$ and $\mathbf{z}$. For this work, $X$ is referred as *treatment variable*, $Y$ as *mediating variable* and $Z$ as *adjustment variables* accounting for confounding effects. With an abuse of notation, features matrices are defined by stacking feature maps along the columns, i.e $\Phi_{\mathbf{x}} := [\phi_x(x_1), ..., \phi_x(x_N)]$. We denote the Gram matrix as $K_{\mathbf{xx}} := \Phi_{\mathbf{x}}^\top \Phi_{\mathbf{x}}$ and the vector of evaluations $k_{x\mathbf{x}}$ as $[k_x(x, x_1), ..., k_x(x, x_N)]$. We define $\Phi_{\mathbf{y}}, \Phi_{\mathbf{z}}$ analogously for $\mathbf{y}$ and $\mathbf{z}$.

Lastly, we denote $T = f(Y) + \epsilon$ as our *target variable*, which is modelled as some noisy evaluation of a function $f : \mathcal{Y} \to \mathcal{T}$ on $Y$ while $\epsilon$ being some random noise. For our problem setup we observe a second dataset of i.i.d realisations $\mathcal{D}_2 = \{\tilde{y}_j, t_j\}_{j=1}^M$ from the joint $\mathbb{P}_{YT}$ independent of $\mathcal{D}_1$. Again, we define $\tilde{\mathbf{y}} := [\tilde{y}_1, ..., \tilde{y}_M]^\top$ and $\mathbf{t} := [t_1, ..., t_M]^\top$ just like for $\mathcal{D}_1$. See Fig.2 for illustration.

## 2 Background

Representing interventional distributions in an RKHS[1] has been explored in different contexts [18, 17, 19]. In particular, when the treatment is continuous, [17] introduced the *Interventional Mean Embeddings* (IMEs) to model densities in an RKHS by utilising smoothness across treatments. Given that IME is an important building block to our contribution, we give it a detailed review by first introducing the key concepts of *do*-calculus [20] and conditional mean embeddings [21].

### 2.1 Interventional distribution and *do*-calculus

In this work, we consider the structural causal model [20] (SCM) framework, where a causal directed acyclic graph (DAG) $\mathcal{G}$ is given and encodes knowledge of existing causal mechanisms amongst the variables in terms of conditional independencies. Given random variables $X$ and $Y$, a central question in interventional inference [20] is to estimate the distribution $p(Y|do(X) = x)$, where $\{do(X) = x\}$ represents an intervention on $X$ whose value is set to $x$. Note that this quantity is not directly observed given that we are usually only given observational data, i.e, data sampled from the conditional $p(Y|X)$ but not from the interventional density $p(Y|do(X))$. However, Pearl [20] developed *do*-calculus which allows us to estimate interventional distributions from purely observational distributions under the identifiability assumption. Here we present the backdoor and front-door adjustments, which are the fundamental components of DAG based causal inference.

The backdoor adjustment is applicable when there are observed confounding variables $Z$ between the cause $X$ and the effect $Y$ (see Fig. 3 (Top)). In order to correct for this confounding bias we can use the following equation, adjusting for $Z$ as $p(Y|do(X) = x) = \int_{\mathcal{Z}} p(Y|X = x, z)p(z)dz$.

The front-door adjustment applies to cases when confounders are unobserved (see Fig. 3 (Bottom)). Given a set of front-door adjustment variables $Z$, we can again correct the estimate for the causal effect from $X$ to $Y$ with $p(Y|do(X) = x) = \int_{\mathcal{Z}} \int_{\mathcal{X}} p(Y|x', z)p(z|X = x)p(x')dx'dz$.

We rewrite the above formulae in a more general form as we show below. For the remainder of the paper we will opt for this notation:

$$p(Y|do(X) = x) = \mathbb{E}_{\Omega_x}[p(Y|\Omega_x)] = \int p(Y|\Omega_x)p(\Omega_x)d\Omega_x \quad (1)$$

Figure 3: (Top) Backdoor adjustment (Bottom) Front-door adjustment, dashed edges denote unobserved confounders.

For backdoor we have $\Omega_x = \{X = x, Z\}$ and $p(\Omega_x) = \delta_x p(Z)$ where $\delta_x$ is the Dirac measure at $X = x$. For front-door, $\Omega_x = \{X', Z\}$ and $p(\Omega_x) = p(X')p(Z|X = x)$.

---

[1]We refer the reader to a detailed review of RKHS methods provided in the Appendix of [5]

## 2.2 Conditional Mean Embeddings

Kernel mean embeddings of distributions provide a powerful framework for representing probability distributions [11, 21] in an RKHS. In particular, we work with conditional mean embeddings (CMEs) in this paper. Given random variables $X, Y$ with joint distribution $\mathbb{P}_{XY}$, the conditional mean embedding with respect to the conditional density $p(Y|X = x)$, is defined as:

$$\mu_{Y|X=x} := \mathbb{E}_{Y|X=x}[\phi_y(Y)] = \int_{\mathcal{Y}} \phi_y(y)p(y|X=x)dy \tag{2}$$

CMEs allow us to represent the distribution $p(Y|X = x)$ as an element $\mu_{Y|X=x}$ in the RKHS $\mathcal{H}_{k_y}$ without having to model the densities explicitly. Following [21], CMEs can be associated with a Hilbert-Schmidt operator $\mathcal{C}_{Y|X} : \mathcal{H}_{k_x} \to \mathcal{H}_{k_y}$, known as the conditional mean embedding operator, which satisfies $\mu_{Y|X=x} = \mathcal{C}_{Y|X}\phi_x(x)$ where $\mathcal{C}_{Y|X} := \mathcal{C}_{YX}\mathcal{C}_{XX}^{-1}$ with $\mathcal{C}_{YX} := \mathbb{E}_{Y,X}[\phi_y(Y) \otimes \phi_x(X)]$ and $\mathcal{C}_{XX} := \mathbb{E}_{X,X}[\phi_x(X) \otimes \phi_x(X)]$ being the covariance operators. As a result, the finite sample estimator of $\mathcal{C}_{Y|X}$ based on the dataset $\{\mathbf{x}, \mathbf{y}\}$ can be written as:

$$\hat{\mathcal{C}}_{Y|X} = \Phi_{\mathbf{y}}(K_{\mathbf{xx}} + \lambda I)^{-1}\Phi_{\mathbf{x}}^T \tag{3}$$

where $\lambda > 0$ is a regularization parameter. Note that from Eq.3, [22] showed that the CME can be interpret as a vector-valued kernel ridge regressor (V-KRR) i.e. $\phi_x(x)$ is regressed to an element in $\mathcal{H}_{k_y}$. This is crucial as CMEs allow us to turn the integration, in Eq.2, into a regression task and hence remove the need for explicit density estimation. This insight is important as it allows us to derive analytic forms for our algorithms. Furthermore, the regression formalism of CMEs motivated us to derive a Bayesian version of CME using vector-valued Gaussian Processes (V-GP), see Sec.3.

## 2.3 Interventional Mean Embeddings

*Interventional Mean Embeddings* (IME) [17] combine the above ideas to represent interventional distributions in RKHSs. We derive the front-door adjustment embedding here but the backdoor adjustment follows analogously. Denote $\mu_{Y|do(X)=x}$ as the IME corresponding to the interventional distribution $p(Y|do(X) = x)$, which can be written as:

$$\mu_{Y|do(X)=x} := \int_{\mathcal{Y}} \phi_y(y)p(y|do(X)=x)dy = \int_{\mathcal{X}} \int_{\mathcal{Z}} \underbrace{\left(\int_{\mathcal{Y}} \phi_y(y)p(y|x',z)dy\right)}_{\text{CME } \mu_{Y|X=x,Z=z}} p(z|x)p(x')dzdx'$$

using the front-door formula with adjustment variable $Z$, and rearranging the integrals. By definition of CME $\int \phi_y(y)p(y|x',z)dy = C_{Y|X,Z}(\phi_x(x') \otimes \phi_z(z))$ and linearity of integration, we have

$$= C_{Y|X,Z}\left(\underbrace{\int_{\mathcal{X}} \phi_x(x')p(x')dx'}_{=\mu_X} \otimes \underbrace{\int_{\mathcal{Z}} \phi_z(z)p(z|x)dz}_{=\mu_{Z|X=x}}\right) = C_{Y|X,Z}\left(\mu_X \otimes \mu_{Z|X=x}\right)$$

Using notations from Sec.2.1, embedding interventional distributions into an RKHS is as follows.

**Proposition 1.** *Given an identifiable do-density of the form $p(Y|do(X) = x) = \mathbb{E}_{\Omega_x}[p(Y|\Omega_x)]$, the general form of the empirical interventional mean embedding is given by,*

$$\hat{\mu}_{Y|do(X)=x} = \Phi_Y(K_{\Omega_x} + \lambda I)^{-1}\Phi_{\Omega_x}(x)^{\top} \tag{4}$$

*where $K_{\Omega_x} = K_{XX} \odot K_{ZZ}$ and $\Phi_{\Omega_x}(x)$ is derived depending on $p(\Omega_x)$. In particular, for backdoor adjustments, $\Phi_{\Omega_x}^{(bd)}(x) = \Phi_X^{\top} k_X(x,\cdot) \odot \Phi_Z^{\top}\hat{\mu}_z$ and for front-door $\Phi_{\Omega_x}^{(fd)}(x) = \Phi_X^{\top}\hat{\mu}_X \odot \Phi_Z^{\top}\hat{\mu}_{Z|X=x}$.*

# 3 Our Proposed Method

**Two-staged Causal Learning.** Given two independent datasets $\mathcal{D}_1 = \{(x_i, z_i, y_i)\}_{i=1}^N$ and $\mathcal{D}_2 = \{(\tilde{y}_j, t_j)\}_{j=1}^M$, our goal is to model the average treatment effect in $T$ when intervening on variable $X$, i.e model $g(x) = \mathbb{E}[T|do(X) = x]$. Note that the target variable $T$ and the treatment variable $X$ are never jointly observed. Rather, they are linked via a mediating variable $Y$ observed in both datasets.

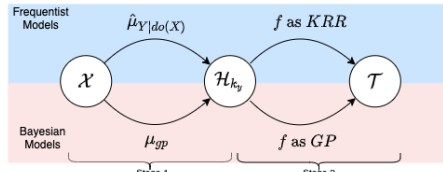

Figure 4: Two-staged causal learning problem

| METHODS | Stage 1 | Stage 2 |
|---|---|---|
| IME [17] | KRR | KRR |
| IMP (Ours) | KRR | GP |
| BAYESIME (Ours) | GP | KRR |
| BAYESIMP (Ours) | GP | GP |

Table 1: Summary of our proposed methods

In our problem setting, we make the following two assumptions: **(A1)** The treatment only affects the target through the mediating variable, i.e $T \perp\!\!\!\perp do(X)|Y \to P(T|do(X), Y) = p(T|Y)$, in other words, that in the true data generating model P(X, Y, Z, T), all causal paths from X to T are mediated through Y. and **(A2)** Function $f$ given by $f(y) = \mathbb{E}[T|Y = y]$ belongs to an RKHS $\mathcal{H}_{k_y}$.[2]

We can thus express the average treatment effect as:

$$g(x) = \mathbb{E}[T|do(X) = x] = \int_{\mathcal{Y}} \underbrace{\mathbb{E}[T|do(X) = x, Y = y]}_{=\mathbb{E}[T|Y=y], \text{ since } T \perp\!\!\!\perp do(X)|Y} p(y|do(X) = x)dy \quad (5)$$

$$= \int_{\mathcal{Y}} f(y)p(y|do(X) = x)dy = \langle f, \mu_{Y|do(X)=x}\rangle_{\mathcal{H}_{k_y}}. \quad (6)$$

The final expression decomposes the problem of estimating $g$ into that of estimating the IME $\mu_{Y|do(X)}$ (which can be done using $\mathcal{D}_1$) and that of estimating the integrand $f : \mathcal{Y} \to \mathcal{T}$ (which can be done using $\mathcal{D}_2$). Each of these two components can either be estimated using a GP or KRR approach (See Table 1). Furthermore, the reformulation as an RKHS inner product is crucial, as it circumvents the need for density estimation as well as the need for subsequent integration in Eq.6. Rather, the main parts of the task can now be viewed as two instances of regression (recall that mean embeddings can be viewed as vector-valued regression).

To model $g$ and quantify its uncertainty, we propose 3 GP-based approaches. While the first 2 methods, *Interventional Mean Process* (IMP) and *Bayesian Interventional Mean Embedding* (BAYESIME) are novel derivations that allow us to quantify uncertainty from either one of the datasets, we treat them as intermediate yet necessary steps to derive our main algorithm, *Bayesian Interventional Mean Process* (BAYESIMP), which allows us to quantify uncertainty from both sources in a principled way. For a summary of the methods, see Fig.4 and Table 1. All derivations are included in the appendix.

***Interventional Mean Process***: Firstly, we train $f$ as a GP using $\mathcal{D}_2$ and model $\mu_{Y|do(X)=x}$ as V-KRR using $\mathcal{D}_1$. By drawing parallels to Bayesian quadrature [24] and conditional mean process introduced in [25], the integral of interest $g(x) = \int f(y)p(y|do(X) = x)dy$ will be a GP indexed by the treatment variable $X$. We can then use the empirical embedding $\hat{\mu}_{Y|do(X)}$ learnt in $\mathcal{D}_1$ to obtain an analytic mean and covariance of $g$.

***Bayesian Interventional Mean Embedding***: Next, to account for the uncertainty from $\mathcal{D}_1$, we model $f$ as a KRR and $\mu_{Y|do(X)=x}$ using a V-GP. We introduce our novel *Bayesian Learning of Conditional Mean Embeddings* (BAYESCME), which uses a *nuclear dominant kernel* [26] construction, similar to [27], to ensure that the inner product $\langle f, \mu_{Y|do(X)=x}\rangle$ is well-defined. As the embedding is a GP, the resulting inner product is also a GP and hence takes into account the uncertainty in $\mathcal{D}_1$.(See Prop. 4).

***Bayesian Interventional Mean Process***: Lastly, in order to account for uncertainties coming from both $\mathcal{D}_1$ and $\mathcal{D}_2$, we combine ideas from the above IMP and BAYESIME. We place GPs on both $f$ and $\mu_{Y|do(X)}$ and use their inner product to model $\mathbb{E}[T|do(X)]$. Interestingly, the resulting uncertainty can be interpreted as the sum of uncertainties coming from IMP and BAYESIME with an additional interaction term (See Prop.5).

## 3.1 Interventional Mean Process

Firstly, we consider the case where $f$ is modelled using a GP and $\mu_{Y|do(X)=x}$ using a V-KRR. This allows us to take into account the uncertainty from $\mathcal{D}_2$ by modelling the relationship between $Y$ and

---

[2]We note that this two-staged setup resembles Instrumental Variable [23] (IV) regression. However, our general setup allows the IV and the treatment to be confounded, which is not the case in standard IV regression.

$T$ using in a GP. Drawing parallels to Bayesian quadrature [24] where integrating $f$ with respect to a marginal measure results into a Gaussian random variable, we integrate $f$ with respect to a conditional measure, thus resulting in a GP indexed by the conditioning variable. Note that [25] studied this GP in a non-causal setting, for a very specific downscaling problem. In this work, we extend their approach to model uncertainty in the causal setting. The resulting mean and covariance are then estimated analytically, i.e without integrals, using the empirical IME $\hat{\mu}_{Y|do(X)}$ learnt from $\mathcal{D}_1$, see Prop.2.

**Proposition 2** (IMP). *Given dataset* $D_1 = \{(x_i, y_i, z_i)\}_{i=1}^{N}$ *and* $D_2 = \{(\tilde{y}_j, t_j)\}_{j=1}^{M}$, *if $f$ is the posterior GP learnt from* $\mathcal{D}_2$, *then* $g = \int f(y) p(y|do(X)) dy$ *is a GP* $\mathcal{GP}(m_1, \kappa_1)$ *defined on the treatment variable $X$ with the following mean and covariance estimated using* $\hat{\mu}_{Y|do(X)}$,

$$m_1(x) = \langle \hat{\mu}_{Y|do(x)}, m_f \rangle_{\mathcal{H}_{k_y}} = \Phi_{\Omega_x}(x)^{\top} (K_{\Omega_x} + \lambda I)^{-1} K_{\mathbf{y}\tilde{\mathbf{y}}} (K_{\tilde{\mathbf{y}}\tilde{\mathbf{y}}} + \lambda_f I)^{-1} \mathbf{t} \tag{7}$$

$$\kappa_1(x, x') = \hat{\mu}_{Y|do(x)}^{\top} \hat{\mu}_{Y|do(x')} - \hat{\mu}_{Y|do(x)}^{\top} \Phi_{\tilde{\mathbf{y}}} (K_{\tilde{\mathbf{y}}\tilde{\mathbf{y}}} + \lambda I)^{-1} \Phi_{\tilde{\mathbf{y}}}^{\top} \hat{\mu}_{Y|do(x')} \tag{8}$$

$$= \Phi_{\Omega_x}(x)^{\top} (K_{\Omega_x} + \lambda I)^{-1} \tilde{K}_{\mathbf{yy}} (K_{\Omega_x} + \lambda I)^{-1} \Phi_{\Omega_x}(x') \tag{9}$$

*where* $\hat{\mu}_{Y|do(x)} = \hat{\mu}_{Y|do(X)=x}, K_{\tilde{\mathbf{y}}\mathbf{y}} = \Phi_{\tilde{\mathbf{y}}}^{\top} \Phi_{\mathbf{y}}$, $m_f$ *and* $\tilde{K}_{\mathbf{yy}}$ *are the posterior mean function and covariance of $f$ evaluated at $\mathbf{y}$ respectively.* $\lambda > 0$ *is the regularisation of the* CME. $\lambda_f > 0$ *is the noise term for GP $f$.* $\Omega_x$ *is the set of variables as specified in Prop.1.*

**Summary:** The posterior covariance between $x$ and $x'$ in IMP can be interpreted as the similarity between their corresponding empirical IMEs $\hat{\mu}_{Y|do(X)=x}$ and $\hat{\mu}_{Y|do(X)=x'}$ weighted by the posterior covariance $\tilde{K}_{\mathbf{yy}}$, where the latter corresponds to the uncertainty when modelling $f$ as a GP in $\mathcal{D}_2$. However, since $f$ only considers uncertainty in $\mathcal{D}_2$, we need to develop a method that allows us to quantify uncertainty when learning the IME from $\mathcal{D}_1$. In the next section, we introduce a Bayesian version of CME, which then lead to BAYESIME, a remedy to this problem.

## 3.2 Bayesian Interventional Mean Embedding

To account for the uncertainty in $\mathcal{D}_1$ when estimating $\mu_{Y|do(X)}$, we consider a GP model for CME, and later extend to the interventional embedding IME. We note that Bayesian formulation of CMEs has also been considered in [28], but with a specific focus on discrete target spaces.

**Bayesian learning of conditional mean embeddings with V-GP.** As mentioned in Sec.2, CMEs have a clear "feature-to-feature" regression perspective, i.e $\mathbb{E}[\phi_y(Y)|X = x]$ is the result of regressing $\phi_y(Y)$ onto $\phi_x(X)$. Hence, we consider a vector-valued GP construction to estimate the CME.

Let $\mu_{gp}(x, y)$ be a GP that models $\mu_{Y|X=x}(y)$. Given that $f \in \mathcal{H}_{k_y}$, for $\langle f, \mu_{gp}(x, \cdot) \rangle_{\mathcal{H}_{k_y}}$ to be well defined, we need to ensure $\mu_{gp}(x, \cdot)$ is also restricted to $\mathcal{H}_{k_y}$ for any fixed $x$. Consequently, we cannot define a $\mathcal{GP}(0, k_x \otimes k_y)$ prior on $\mu_{gp}$ as usual, as draws from such prior will almost surely fall outside $\mathcal{H}_{k_x} \otimes \mathcal{H}_{k_y}$ [26]. Instead we define a prior over $\mu_{gp} \sim \mathcal{GP}(0, k_x \otimes r_y)$, where $r_y$ is a *nuclear dominant kernel* [26] over $k_y$, which ensures that samples paths of $\mu_{gp}$ live in $\mathcal{H}_{k_x} \otimes \mathcal{H}_{k_y}$ almost surely. In particular, we follow a similar construction as in [27] and model $r_y$ as $r_y(y_i, y_j) = \int k_y(y_i, u) k_y(u, y_j) \nu(du)$ where $\nu$ is some finite measure on $Y$. Hence we can now setup a vector-valued regression in $\mathcal{H}_{k_y}$ as follows:

$$\phi_y(y_i) = \mu_{gp}(x_i, \cdot) + \lambda^{\frac{1}{2}} \epsilon_i \tag{10}$$

where $\epsilon_i \sim \mathcal{GP}(0, r)$ are independent noise functions. By taking the inner product with $\phi_y(y')$ on both sides, we then obtain $k_y(y_i, y') = \mu_{gp}(x_i, y') + \lambda^{\frac{1}{2}} \epsilon_i(y')$. Hence, we can treat $k(y_i, y_j)$ as noisy evaluations of $\mu_{gp}(x_i, y_j)$ and obtain the following posterior mean and covariance for $\mu_{gp}$.

**Proposition 3** (BAYESCME). *The posterior GP of $\mu_{gp}$ given observations $\{\mathbf{x}, \mathbf{y}\}$ has the following mean and covariance:*

$$m_{\mu}((x, y)) = k_{x\mathbf{x}} (K_{\mathbf{xx}} + \lambda I)^{-1} K_{\mathbf{yy}} R_{\mathbf{yy}}^{-1} r_{\mathbf{y}y} \tag{11}$$

$$\kappa_{\mu}((x, y), (x', y')) = k_{xx'} r_{y,y'} - k_{x\mathbf{x}} (K_{\mathbf{xx}} + \lambda I)^{-1} k_{\mathbf{xx'}} r_{y\mathbf{y}} R_{\mathbf{yy}}^{-1} r_{\mathbf{y}y'} \tag{12}$$

*In addition, the following marginal likelihood can be used for hyperparameter optimisation,*

$$-\frac{N}{2} \Big( \log |K_{\mathbf{xx}} + \lambda I| + \log |R| \Big) - \frac{1}{2} \text{Tr} \Big( (K_{\mathbf{xx}} + \lambda I)^{-1} K_{\mathbf{yy}} R_{\mathbf{yy}}^{-1} K_{\mathbf{yy}} \Big) \tag{13}$$

Note that in practice we fix the lengthscale of $k_y$ and $r_y$ when optimising the above likelihood. This is to avoid trivial solutions for the vector-valued regression problem as discussed in [10]. The Bayesian version of the IME is derived analogously and we refer the reader to appendix due to limited space.

Finally, with V-GPs on embeddings defined, we can model $g(x)$ as $\langle f, \mu_{gp}(x, \cdot)\rangle_{H_{k_y}}$, which due to the linearity of the inner product, is itself a GP. Here, we first considered the case where $f$ is a KRR learnt from $\mathcal{D}_2$ and call the model BAYESIME.

**Proposition 4** (BAYESIME). *Given dataset $D_1 = \{(x_i, y_i, z_i)\}_{i=1}^N$ and $D_2 = \{(\tilde{y}_j, t_j)\}_{j=1}^M$, if $f$ is a KRR learnt from $\mathcal{D}_2$ and $\mu_{Y|do(X)}$ modelled as a V-GP using $\mathcal{D}_1$, then $g = \langle f, \mu_{Y|do(X)}\rangle \sim \mathcal{GP}(m_2, \kappa_2)$ where,*

$$m_2(x) = \Phi_{\Omega_x}(x)^\top (K_{\Omega_x} + \lambda I)^{-1} K_{\mathbf{y}\hat{\mathbf{y}}} R_{\mathbf{yy}}^{-1} R_{\mathbf{y}\tilde{\mathbf{y}}} A \tag{14}$$

$$\kappa_2(x, x') = B\Phi_{\Omega_x}(x)^\top \Phi_{\Omega_x}(x) - C\Phi_{\Omega_x}(x)^\top (K_{\Omega_x} + \lambda I)^{-1} \Phi_{\Omega_x}(x') \tag{15}$$

*where $A = (K_{\tilde{\mathbf{y}}\tilde{\mathbf{y}}} + \lambda_f I)^{-1}\mathbf{t}$, $B = A^\top R_{\tilde{\mathbf{y}}\tilde{\mathbf{y}}} A$ and $C = A^\top R_{\tilde{\mathbf{y}}\mathbf{y}} R_{\mathbf{yy}}^{-1} R_{\mathbf{y}\tilde{\mathbf{y}}} A$*

**Summary:** Constants $B$ and $C$ in $\kappa_2$ can be interpreted as different estimation of $||f||_{\mathcal{H}_{k_y}}$, i.e the RKHS norm of $f$. As a result, for problems that are "harder" to learn in $\mathcal{D}_2$, i.e. corresponding to larger magnitude of $||f||_{\mathcal{H}_{k_y}}$, will result into larger values of $B$ and $C$. Therefore the covariance $\kappa_2$ can be interpreted as uncertainty in $\mathcal{D}_1$ scaled by the difficulty of the problem to learn in $\mathcal{D}_2$.

### 3.3 Bayesian Interventional Mean Process

To incorporate both uncertainties in $\mathcal{D}_1$ and $\mathcal{D}_2$, we combine ideas from IMP and BAYESIME to estimate $g = \langle f, \mu_{Y|do(X)}\rangle$ by placing GPs on both $f$ and $\mu_{Y|do(X)}$. Again as before, a nuclear dominant kernel $r_y$ was used to ensure the GP $f$ is supported on $\mathcal{H}_{k_y}$. For ease of computation, we consider a finite dimensional approximation of the GPs $f$ and $\mu_{Y|do(X)}$ and estimate $g$ as the RKHS inner product between them. In the following we collate $\mathbf{y}$ and $\tilde{\mathbf{y}}$ into a single set of points $\hat{\mathbf{y}}$, which can be seen as landmark points for the finite approximation [29]. We justify this in the Appendix.

**Proposition 5** (BAYESIMP). *Let $f$ and $\mu_{Y|do(X)}$ be GPs learnt as above. Denote $\tilde{f}$ and $\tilde{\mu}_{Y|do(X)}$ as the finite dimensional approximation of $f$ and $\mu_{Y|do(X)}$ respectively. Then $\tilde{g} = \langle \tilde{f}, \tilde{\mu}_{Y|do(X)}\rangle$ has the following mean and covariance:*

$$m_3(x) = E_x K_{\mathbf{y}\hat{\mathbf{y}}} K_{\hat{\mathbf{y}}\hat{\mathbf{y}}}^{-1} R_{\hat{\mathbf{y}}\tilde{\mathbf{y}}} (R_{\tilde{\mathbf{y}}\tilde{\mathbf{y}}} + \lambda_f I)^{-1}\mathbf{t} \tag{16}$$

$$\kappa_3(x, x') = \underbrace{E_x \Theta_1^\top \tilde{R}_{\hat{\mathbf{y}}\hat{\mathbf{y}}} \Theta_1 E_{x'}^\top}_{\text{Uncertainty from } \mathcal{D}_1} + \underbrace{\Theta_2^{(a)} F_{xx'} - \Theta_2^{(b)} G_{xx'}}_{\text{Uncertainty from } \mathcal{D}_2} + \underbrace{\Theta_3^{(a)} F_{xx'} - \Theta_3^{(b)} G_{xx'}}_{\text{Uncertainty from Interaction}} \tag{17}$$

*where $E_x = \Phi_{\Omega_x}(x)^\top (K_{\Omega_x} + \lambda I)^{-1}$, $F_{xx'} = \Phi_{\Omega_x}(x)^\top \Phi_{\Omega_x}(x')$, $G_{xx'} = \Phi_{\Omega_x}(x)^\top (K_{\Omega_x} + \lambda I)^{-1} \Phi_{\Omega_x}(x')$, and $\Theta_1 = K_{\hat{\mathbf{y}}\hat{\mathbf{y}}}^{-1} R_{\hat{\mathbf{y}}\mathbf{y}} R_{\mathbf{yy}}^{-1} K_{\mathbf{yy}}$, $\Theta_2^{(a)} = \Theta_4^\top R_{\hat{\mathbf{y}}\hat{\mathbf{y}}} \Theta_4$, $\Theta_2^{(b)} = \Theta_4^\top R_{\hat{\mathbf{y}}\mathbf{y}} R_{\mathbf{yy}}^{-1} R_{\mathbf{y}\hat{\mathbf{y}}} \Theta_4$ and $\Theta_3^{(a)} = tr(K_{\hat{\mathbf{y}}\hat{\mathbf{y}}}^{-1} R_{\hat{\mathbf{y}}\hat{\mathbf{y}}} K_{\hat{\mathbf{y}}\hat{\mathbf{y}}}^{-1} \bar{R}_{\hat{\mathbf{y}}\hat{\mathbf{y}}})$, $\Theta_3^{(b)} = tr(R_{\hat{\mathbf{y}}\mathbf{y}} R_{\mathbf{yy}}^{-1} R_{\mathbf{y}\hat{\mathbf{y}}} K_{\hat{\mathbf{y}}\hat{\mathbf{y}}}^{-1} \bar{R}_{\hat{\mathbf{y}}\hat{\mathbf{y}}} K_{\hat{\mathbf{y}}\hat{\mathbf{y}}}^{-1})$ and $\Theta_4 = K_{\hat{\mathbf{y}}\hat{\mathbf{y}}}^{-1} R_{\hat{\mathbf{y}}\tilde{\mathbf{y}}} (K_{\tilde{\mathbf{y}}\tilde{\mathbf{y}}} + \lambda_f)^{-1}\mathbf{t}$. $\bar{R}_{\hat{\mathbf{y}}\hat{\mathbf{y}}}$ is the posterior covariance of $f$ evaluated at $\hat{\mathbf{y}}$*

**Summary:** While the first two terms in $\kappa_3$ resemble the uncertainty estimates from IMP and BAYESIME, the last term acts as an extra interaction between the two uncertainties from $\mathcal{D}_1$ and $\mathcal{D}_2$. We note that unlike IMP and BAYESIME, $\tilde{g}$ from Prop.5 is not a GP as inner products between Gaussian vectors are not Gaussian. Nonetheless, the mean and covariance can be estimated.

## 4 Experiments

In this section, we first present an ablation studies on how our methods would perform under settings where we have missing data parts at different regions of the two datasets. We then demonstrate BAYESIMP's proficiency in the Causal Bayesian Optimisation setting.

In particular, we compare our methods against the sampling approach considered in [15]. [15] start by modelling $f : Y \rightarrow T$ as GP and estimate the density $p(Y|do(X))$ using a GP along with *do*-calculus. Then given a treatment $x$, we obtain $L$ samples of $y_l$ and $R$ samples of $f_r$ from their posterior GPs.

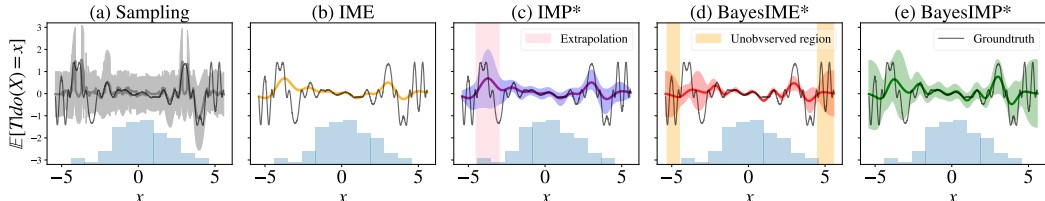

Figure 5: Ablation studies of various methods in estimating uncertainties for an illustrative experiment. $*$ indicates our methods. $N = M = 100$ data points are used. Uncertainty from sampling gives a uniform estimate of uncertainty and IME does not come with uncertainty estimates. We see IMP and BAYESIME covering different regions of uncertainty while BAYESIMP takes the best of both worlds.

The empirical mean and standard deviation of the samples $\{f_r(y_l)\}_{l=1,r=1}^{L,R}$ can now be taken to estimate $\mathbb{E}[T|do(X) = x]$ as well as the correspondingly uncertainty. We emphasize that this point estimation requires repeated sampling and is thus inefficient compared to our approaches, where we explicitly model the uncertainty as covariance function.

**Ablation study.** In order to get a better intuition into our methods, we will start off with a preliminary example, where we investigate the uncertainty estimates in a toy case. We assume two simple causal graphs $X \to Y$ for $\mathcal{D}_1$ and $Y \to T$ for $\mathcal{D}_2$ and the goal is to estimate $\mathbb{E}[T|do(X) = x]$ (generating process given in the appendix). We compare our methods from Sec.3 with the sampling-based uncertainty estimation approach described above. In Fig.5 we plot the mean and the $95\%$ credible interval of the resulting GP models for $\mathbb{E}[T|do(X) = x]$. On the $x$-axis we also plotted a histogram of the treatment variable $x$ to illustrate its density.

From Fig.5(a), we see that the uncertainty for sampling is rather uniform across the ranges of $x$ despite the fact we have more data around $x = 0$. This is contrary to our methods, which show a reduction of uncertainty at high $x$ density regions. In particular, $x = -5$ corresponds to an extrapolation of data, where $x$ gets mapped to a region of $y$ where there is no data in $\mathcal{D}_2$. This fact is nicely captured by the spike of credible interval in Fig.5(c) since IMP utilises uncertainty from $\mathcal{D}_2$ directly. Nonetheless, IMP failed to capture the uncertainty stemming from $\mathcal{D}_1$, as seen from the fact that the credible interval did not increase as we have less data in the region $|x| > 5$. In contrast, BAYESIME (Fig.5(d)) gives higher uncertainty around low $x$ density regions but failed to capture the **extrapolation** phenomenon. Finally, BAYESIMP Fig.5(e) seems to inherit the desirable characteristics from both IMP and BAYESIME, due to taking into account uncertainties from both $\mathcal{D}_1, \mathcal{D}_2$. Hence, in the our experiments, we focus on BAYESIMP and refer the reader to the appendix for the remaining methods.

**BayesIMP for Bayesian Optimisation (BO).** We now demonstrate, on both synthetic and real-world data, the usefulness of the uncertainty estimates obtained using our methods in BO tasks. Our goal is to utilise the uncertainty estimates to direct the search for the optimal value of $\mathbb{E}[T|do(X) = x]$ by querying as few values of the treatment variable $X$ as possible, i.e. we want to optimize for $x^* = \arg\min_{x \in \mathcal{X}} \mathbb{E}[T|do(X) = x]$. For the first synthetic experiment (see Fig.6 (Top)), we will use the following two datasets: $\mathcal{D}_1 = \{x_i, u_i, z_i, y_i\}_{i=1}^N$ and $\mathcal{D}_2 = \{\tilde{y}_j, t_j\}_{j=1}^M$. Note that BAYESIMP from Prop.5 is not a GP as inner products between Gaussian vectors are not Gaussian. Nonetheless, with the mean and covariance estimated, we will use moment matching to construct a GP out of BAYESIMP for posterior inference. At the start, we are given $\mathcal{D}_1$ and $\mathcal{D}_2$, where these observations are used to construct a GP prior for the interventional effect of $X$ on $T$, i.e $\mathbb{E}[T|do(X) = x]$, to provide a "warm" start for the BO.

Again we compare BAYESIMP with the sampling-based estimation of $\mathbb{E}[T|do(X)]$ and its uncertainty, which is exactly the formulation used in the Causal Bayesian Optimisation algorithm (CBO) [15]. In order to demonstrate how BAYESIMP performs in the multimodal setting, we will be considering the case where we have the following distribution on $Y$ i.e. $p(y|u, z) = \pi p_1(y|u, z) + (1 - \pi)p_2(y|u, z)$ where $Y$ is a mixture and $\pi \in [0, 1]$. These scenarios might arise when there is an unobserved binary variable which induces a switching between two regimes on how $Y$ depends on $(U, Z)$. In this case, the GP model of [15] would only capture the conditional expecta-

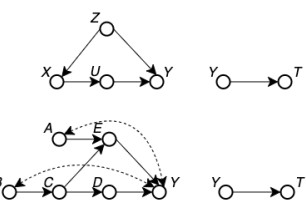

Figure 6: Illustration of synthetic data experiments.

tion of $Y$ with an inflated variance, leading to slower convergence and higher variance in the estimates of the prior as we will show in our experiments. Throughout our experiments, similarly to [15], we will be using the expected improvement (EI) acquisition function to select the next point to query.

**Synthetic data experiments.** We compare BAYESIMP to CBO as well as to a simple GP with no learnt prior as baseline. We will be using $N = 100$ datapoints for $\mathcal{D}_1$ and $M = 50$ datapoints $\mathcal{D}_2$. We ran each method 10 times and plot the resulting standard deviation for each iteration in the figures below. The data generation and details were added in the Appendix. We see from the Fig.7 that

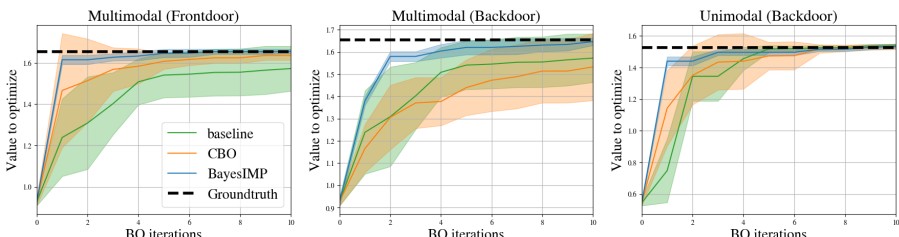

Figure 7: We are interested in finding the maximal value of $\mathbb{E}[T|do(X) = x]$ with as few BO iterations as possible. We ran experiments with **multimodality** in $Y$. (Left) Using front-door adjustment (Middle) Using backdoor adjustment (Right) Using backdoor adjustment (**unimodal** $Y$)

BAYESIMP is able to find the maxima much faster and with smaller standard deviations, than the current state-of-the-art method, CBO, using both front-door and backdoor adjustments (Fig.7(Right, Middle)). Given that our method uses more flexible representations of conditional distributions, we are able to circumvent the multimodality problem in $Y$. In addition, we also consider the unimodal version, i.e. $\pi = 0$ (see right Fig.7). We see that the performance of CBO improves in the unimodal setting, however BAYESIMP still converges faster than CBO even in this scenario.

Next, we consider a harder causal graph (see Fig.6 (Bottom)), previously considered in [15]. We again introduce multimodality in the $Y$ variable in order to explore the case of more challenging conditional densities. We see from Fig.8 (Left, Middle), that BAYESIMP again converges much faster to the true optima than CBO [15] and the standard GP prior baseline. We note that the fast convergence of BAYESIMP throughout our experiments is not due to simplicity of the underlying BO problems. Indeed, the BO with a standard GP prior requires significantly more iterations. It is rather the availability of the observational data, allowing us to construct a more appropriate prior, which leads to a "warm" start of the BO procedure.

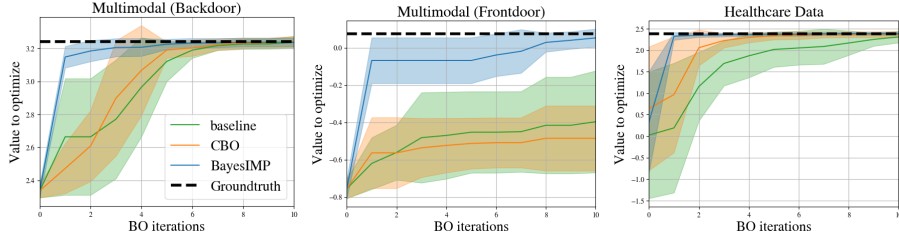

Figure 8: (Left) Experiments where we are interested in $\mathbb{E}[T|do(D) = d]$ with **multimodal** $Y$, (Middle) Experiments where we are interested in $\mathbb{E}[T|do(E) = e]$ with **multimodal** $Y$, (Right) Experiments on **healthcare data** where we are interested in $\mathbb{E}[Cancer\ Volume|do(Statin)]$.

**Healthcare experiments.** We conclude with a healthcare dataset corresponding to our motivating medical example in Fig.1. The causal mechanism graph, also considered in the CBO paper [15], studies the effect of certain drugs (Aspirin/Statin) on Prostate-Specific Antigen (PSA) levels [6]. In our case, we modify *statin* to be continuous, in order to optimize for the correct drug dosage. However, in contrast to [15], we consider a second experimental dataset, arising from a different medical study, which looks into the connection between *PSA* levels and *cancer volume* amount in patients [7]. Similar to the original CBO paper [15], given that interventional data is hard to obtain, we construct data generators based on the true data collected in [7]. This is done by firstly fitting a GP on the data and then sampling from the posterior (see Appendix for more details). Hence this is the

perfect testbed for our model where we are interested in $\mathbb{E}[\textit{Cancer Volume}|\textit{do(Statin)}]$. We see from Fig.8 (Right) that BAYESIMP again converges to the true optima faster than CBO hence allowing us to find the fastest ways of optimizng *cancer volume* by requesting much less interventional data. This could be critical as interventional data in real-life situations can be very expensive to obtain.

## 5 Discussion and Conclusion

In this paper we propose BAYESIMP for quantifying uncertainty in the setting of causal data fusion. In particular, our proposed method BAYESIMP allows us to represent interventional densities in the RKHS without explicit density estimation, while still accounting for epistemic and aleatoric uncertainties. We demonstrated the quality of the uncertainty estimates in a variety of Bayesian optimization experiments, in both synthetic and real-world healthcare datasets, and achieve significant improvement over current SOTA in terms of convergence speed. However, we emphasize that BAYESIMP is not designed to replace CBO but rather an alternative model for interventional effects.

In the future, we would like to improve BAYESIMP over several limitations. As in [15], we assumed full knowledge of the underlying causal graph, which might be limiting in practice. Furthermore, as the current formulation of BAYESIMP only allows combination of two causal graphs, we hope to generalise the algorithm into arbitrary number of graphs in the future. Causal graphs with recurrent structure will be an interesting direction to explore. Lastly, we would also like to include a cost function as in [15] to constrain the search space to the most sensible solutions.

## 6 Acknowledgements

The authors would like to thank Bobby He, Robert Hu, Kaspar Martens, Jake Fawkes and Joost van Amersfoort for helpful comments. SLC and JFT are supported by the EPSRC and MRC through the OxWaSP CDT programme EP/L016710/1. YWT and DS are supported in part by Tencent AI Lab and DS is supported in part by the Alan Turing Institute (EP/N510129/1). YWT's research leading to these results has received funding from the European Research Council under the European Union's Seventh Framework Programme (FP7/2007-2013) ERC grant agreement no. 617071.

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
