## Societal Impact

In this paper we propose a general framework for embedding interventional distributions into a RKHS while accounting for uncertainty in causal datasets. We believe that this is a crucial problem that has not gotten much attention yet, but is nonetheless important for the future of causal inference research. In particular, given that our proposed method allows us to combine datasets from different studies, we envision that this could potentially be used in a variety of scientific areas such as healthcare, drug discovery etc. Finally, the only potential negative impact, would be, when using biased data. Our method relies on knowing the correct causal graph and hence could be misinterpreted when this is not the case.

# A Additional background on backdoor/front-door adjustments

In causal inference, we are often times interested in the interventional distributions i.e $p(y|do(x))$ rather than $p(y|x)$, as the former allows us to account for confounding effects. In order to obtain the interventional density $p(y|do(x))$, we resort to *do*-calculus [20]. Here below we write out the definition for the 2 most crucial formulaes; the front-door and backdoor adjustments, with which we are able to recover the interventional density using only the conditional ones.

## A.1 Back-door Adjustment

The key intuition of back-door adjustments is to find/adjust a set of confounders that are unaffected by the treatment. We can then study the effect of the treatment has to the target.

**Definition 1** (Back-Door). *A set of variables $Z$ satisfies the backdoor criterion relative to an ordered pair of variables $X_i, X_j$ in a DAG G if:*

1. *no node in $Z$ is a descendant of $X_i$; and*

2. *$Z$ blocks every path between $X_i$ and $X_j$ that contains an arrow into $X_i$*

*Similarly, if $X$ and $Y$ are two disjoint subsets of nodes in G, then $Z$ is said satisfy the back-door criterion relative to $(X, Y)$ if it satisfies the criterion relative to any pair $(X_i, X_j)$ such that $X_i \in X$ and $X_j \in Y$*

Now with a given set $Z$ that satisfies the back-door criterion, we apply the backdoor adjustment,

**Theorem 1** (Back-Door Adjustment). *If a set of variables $Z$ satisfies the back-door criterion relative to $(X, Y)$, then the causal effect of $X$ on $Y$ is identifiable and is given by the formula*

$$P(y|do(X) = x) = \int_z p(y|x, z)p(z)dz \tag{18}$$

## A.2 Front-door Adjustment

Front-door adjustment deals with the case where confounders are unobserved and hence the backdoor adjustment is not applicable.

**Definition 2** (Front-door). *A set of variables $Z$ is said to satisfy the front-door criterion relative to an ordered pair of variables $(X, Y)$ if:*

1. *$Z$ intercepts all directed paths from $X$ to $Y$;*

2. *there is no back-door path from $X$ to $Z$; and*

3. *all back-door paths from $Z$ to $Y$ are blocked by $X$*

Again, with an appropriate front-door adjustment set $Z$, we can identify the do density using the front-door adjustment formula.

**Theorem 2** (Front-Door Adjustment). *If $Z$ satisfies the front-door criterion relative to $(X, Y)$ and if $P(x, z) > 0$, then the causal effect of $X$ on $Y$ is identifiable and is given by the formula:*

$$p(y|do(X) = x) = \int_z p(z|x) \int_{x'} p(y|x', z)p(x')dx'dz \tag{19}$$

## B  Derivations

### B.1  CMP Derivation

**Proposition 2.** *Given dataset $D_1 = \{(x_i, y_i, z_i)\}_{i=1}^N$ and $D_2 = \{(\tilde{y}_j, t_j)\}_{j=1}^M$, if $f$ is the posterior GP learnt from $\mathcal{D}_2$, then $g = \int f(y)p(y|do(X))dy$ is a GP $\mathcal{GP}(m_1, \kappa_1)$ defined on the treatment variable $X$ with the following mean and covariance estimated using $\hat{\mu}_{Y|do(X)}$,*

$$m_1(x) = \langle \hat{\mu}_{Y|do(x)}, m_f \rangle_{\mathcal{H}_{k_y}} = \Phi_{\Omega_x}(x)^\top (K_{\Omega_x} + \lambda I)^{-1} K_{\mathbf{y}\tilde{\mathbf{y}}}(K_{\tilde{\mathbf{y}}\tilde{\mathbf{y}}} + \lambda_f I)^{-1}\mathbf{t} \tag{20}$$

$$\kappa_1(x, x') = \hat{\mu}_{Y|do(x)}^\top \hat{\mu}_{Y|do(x')} - \hat{\mu}_{Y|do(x)}^\top \Phi_{\tilde{\mathbf{y}}}(K_{\tilde{\mathbf{y}}\tilde{\mathbf{y}}} + \lambda I)^{-1}\Phi_{\tilde{\mathbf{y}}}^\top \hat{\mu}_{Y|do(x')} \tag{21}$$

$$= \Phi_{\Omega_x}(x)^\top (K_{\Omega_x} + \lambda I)^{-1}\tilde{K}_{\mathbf{y}\mathbf{y}}(K_{\Omega_x} + \lambda I)^{-1}\Phi_{\Omega_x}(x') \tag{22}$$

*where $\hat{\mu}_{Y|do(x)} = \hat{\mu}_{Y|do(X)=x}, K_{\tilde{\mathbf{y}}\mathbf{y}} = \Phi_{\tilde{\mathbf{y}}}^\top \Phi_{\mathbf{y}}$, $m_f$ and $\tilde{K}_{\mathbf{y}\mathbf{y}}$ are the posterior mean function and covariance of $f$ evaluated at $\mathbf{y}$ respectively. $\lambda > 0$ is the regularisation of the CME. $\lambda_f > 0$ is the noise term for GP $f$. $\Omega_x$ is the set of variables as specified in Prop.1.*

*Proof for Proposition 2.* Integral operator preserves Gaussianity under mild conditions (see conditions [25]), therefore

$$g(x) = \int f(y)dP(y|do(X) = x) \tag{23}$$

is also a Gaussian. For a standard GP prior $f \sim GP(0, k_y)$ and data $D_E = \{(\tilde{y}_j, t_j)\}_{j=1}^M$, standard conjugacy results for GPs lead to the posterior GP with mean $\bar{m}(y) = k_{y\tilde{\mathbf{y}}}(K_{\tilde{\mathbf{y}}\tilde{\mathbf{y}}} + \lambda_f I)^{-1}\mathbf{t}$ and covariance $\bar{k}_y(y, y') = k_y(y, y') - k_{y\tilde{\mathbf{y}}}(K_{\tilde{\mathbf{y}}\tilde{\mathbf{y}}} + \lambda_f I)^{-1}k_{\tilde{\mathbf{y}}y}$. Similar to [24], repeated application of Fubini's theorem yields:

$$\mathbb{E}_f[g(x)] = \mathbb{E}_f\left[\int f(y)dP(y|do(X) = x)\right] = \int \mathbb{E}_f[f(y)]dP(y|do(X) = x) \tag{24}$$

$$= \int \bar{m}(y)dP(y|do(X) = x) = \langle \bar{m}, \hat{\mu}_{Y|do(X)=x}\rangle \tag{25}$$

$$cov(g(x), g(x')) = \int\int cov(f(y), f(y'))dP(y|do(X) = x)dP(y'|do(X) = x) \tag{26}$$

$$= \int\int \bar{k}_y(y, y')dP(y|do(x))dP(y'|do(x')) \tag{27}$$

$$= \langle \mu_{Y|do(x)}, \mu_{Y|do(x')}\rangle - \hat{\mu}_{Y|do(x)}^\top \Phi_{\tilde{\mathbf{y}}}(K_{\tilde{\mathbf{y}}\tilde{\mathbf{y}}} + \lambda I)^{-1}\Phi_{\tilde{\mathbf{y}}}^\top \hat{\mu}_{Y|do(x')} \tag{28}$$

$$= \Phi_{\Omega_x}(x)^\top (K_{\Omega_x} + \lambda I)^{-1}\tilde{K}_{\mathbf{y}\mathbf{y}}(K_{\Omega_x} + \lambda I)^{-1}\Phi_{\Omega_x}(x') \tag{29}$$

$$\square$$

### B.2  Choice of Nuclear Dominant Kernel

Recall in section 3.2, we introduced the nuclear dominant kernel $r_y$ to ensure samples of $\mu_{gp} \sim GP(0, k_x \otimes r_y)$ are supported in $\mathcal{H}_{k_x} \otimes \mathcal{H}_{k_y}$ with probability 1. In the following we will present the analytic form of the nuclear dominant kernel we used in this paper, which is the same as the formulation introduced in Appendix A.2 and A.3 of [27]. Pick $k_y$ as the RBF kernel, i.e

$$k_y(y, y') = \exp\left(-\frac{1}{2}(y - y')^\top \Sigma_\theta(y - y')\right) \tag{30}$$

where $\Sigma_\theta$ is covariance matrix for the kernel $k_y$. The nuclear dominant kernel construction from [27] then yield the following expression:

$$r_y(y, y') = \int k_y(y, u)k_y(u, y')\nu(du) \tag{31}$$

where $\nu$ is some finite measure. If we pick $\nu(du) = \exp(\frac{||u||_2^2}{2\eta^2})du$, then we have

$$r_y(y, y') = (2\pi)^{D/2}|2\Sigma_\theta^{-1} + \eta^{-2}I|^{-1/2}\exp\left(-\frac{1}{2}(y - y')^\top (2\Sigma_\theta)^{-1}(y - y')\right) \tag{32}$$

$$\times \exp\left(-\frac{1}{2}\left(\frac{y + y'}{2}\right)^\top \left(\frac{1}{2}\Sigma_\theta + \eta^2 I\right)^{-1}\left(\frac{y + y'}{2}\right)\right) \tag{33}$$

## B.3 BayesCME derivations

**Proposition 3.** *The posterior* GP *of $\mu_{gp}$ given observations $\{\mathbf{x}, \mathbf{y}\}$ has the following mean and covariance:*

$$m_\mu((x,y)) = k_{x\mathbf{x}}(K_{\mathbf{xx}} + \lambda I)^{-1}K_{\mathbf{yy}}R_{\mathbf{yy}}^{-1}r_{\mathbf{y}y} \tag{34}$$

$$\kappa_\mu((x,y),(x',y')) = k_{xx'}r_{y,y'} - k_{x\mathbf{x}}(K_{\mathbf{xx}} + \lambda I)^{-1}k_{\mathbf{x}x'}r_{\mathbf{y}y}R_{\mathbf{yy}}^{-1}r_{\mathbf{y}y'} \tag{35}$$

*In addition, the following marginal likelihood can be used for hyperparameter optimisation,*

$$-\frac{N}{2}\left(\log|K_{\mathbf{xx}} + \lambda I| + \log|R|\right) - \frac{1}{2}\operatorname{tr}\left((K_{\mathbf{xx}} + \lambda I)^{-1}K_{\mathbf{yy}}R_{\mathbf{yy}}^{-1}K_{\mathbf{yy}}\right) \tag{36}$$

*Proof of Proposition 3.* Recall the Bayesian formulation of CME corresponds to the following model,

$$\mu_{gp} \sim GP(0, k_x \otimes r_y),$$
$$k_y(y_i, y') = \mu_{gp}(x_i, y') + \lambda^{1/2}\epsilon_i(y')$$

with $\epsilon_i \sim GP(0, r_y)$ independently across $i$. Now consider $k_y(y_i, y_j)$ as noisy evaluations of $\mu_{gp}(x_i, y_j)$, we have the predictive posterior mean as

$$
\begin{aligned}
\operatorname{vec}(r_{\mathbf{y}y}k_{x\mathbf{x}})^\top (K_{\mathbf{xx}} \otimes R_{\mathbf{yy}} + \lambda I \otimes R_{\mathbf{yy}})^{-1}\operatorname{vec}(K_{\mathbf{yy}}) &= \operatorname{vec}(r_{\mathbf{y}y}k_{x\mathbf{x}})^\top \left((K_{\mathbf{xx}} + \lambda I)^{-1} \otimes R_{\mathbf{yy}}^{-1}\right)\operatorname{vec}(K_{\mathbf{yy}}) \\
&= \operatorname{vec}(r_{\mathbf{y}y}k_{x\mathbf{x}})^\top \operatorname{vec}\left(R_{\mathbf{yy}}^{-1}K_{\mathbf{yy}}(K_{\mathbf{xx}} + \lambda I)^{-1}\right) \\
&= \operatorname{tr}\left(r_{\mathbf{y}y}k_{x\mathbf{x}}(K_{\mathbf{xx}} + \lambda I)^{-1}K_{\mathbf{yy}}R_{\mathbf{yy}}^{-1}\right) \\
&= k_{x\mathbf{x}}(K_{\mathbf{xx}} + \lambda I)^{-1}K_{\mathbf{yy}}R_{\mathbf{yy}}^{-1}r_{\mathbf{y}y}.
\end{aligned}
$$

And the covariance is,

$$
\begin{aligned}
\kappa((x,y),(x',y')) &= k(x,x')r(y,y') - \operatorname{vec}(r_{\mathbf{y}y}k_{x\mathbf{x}})^\top (K_{\mathbf{xx}} \otimes R_{\mathbf{yy}} + \lambda I \otimes R_{\mathbf{yy}})^{-1}\operatorname{vec}(r_{\mathbf{y}y'}k_{x'\mathbf{x}}) \\
&= k(x,x')r(y,y') - \operatorname{vec}(r_{\mathbf{y}y}k_{x\mathbf{x}})^\top \left((K_{\mathbf{xx}} + \lambda I)^{-1} \otimes R_{\mathbf{yy}}^{-1}\right)\operatorname{vec}(r_{\mathbf{y}y'}k_{x'\mathbf{x}}) \\
&= k(x,x')r(y,y') - \operatorname{vec}(r_{\mathbf{y}y}k_{x\mathbf{x}})^\top \operatorname{vec}\left(R_{\mathbf{yy}}^{-1}r_{\mathbf{y}y'}k_{x'\mathbf{x}}(K_{\mathbf{xx}} + \lambda I)^{-1}\right) \\
&= k(x,x')r(y,y') - \operatorname{tr}\left(r_{\mathbf{y}y}k_{x\mathbf{x}}(K_{\mathbf{xx}} + \lambda I)^{-1}k_{\mathbf{x}x'}r_{y'\mathbf{y}}R_{\mathbf{yy}}^{-1}\right) \\
&= k(x,x')r(y,y') - k_{x\mathbf{x}}(K_{\mathbf{xx}} + \lambda I)^{-1}k_{\mathbf{x}x'}r_{y'\mathbf{y}}R_{\mathbf{yy}}^{-1}r_{\mathbf{y}y}.
\end{aligned}
$$

To compute the log likelihood, note that it contains the following two terms:

$$
\begin{aligned}
\operatorname{vec}(K_{\mathbf{yy}})^\top (K_{\mathbf{xx}} \otimes R_{\mathbf{yy}} + \lambda I \otimes R_{\mathbf{yy}})^{-1}\operatorname{vec}(K_{\mathbf{yy}}) &= \operatorname{vec}(K_{\mathbf{yy}})^\top \left((K_{\mathbf{xx}} + \lambda I)^{-1} \otimes R_{\mathbf{yy}}^{-1}\right)\operatorname{vec}(K_{\mathbf{yy}}) \\
&= \operatorname{vec}(K_{\mathbf{yy}})^\top \operatorname{vec}\left(R_{\mathbf{yy}}^{-1}K_{\mathbf{yy}}(K_{\mathbf{xx}} + \lambda I)^{-1}\right) \\
&= \operatorname{tr}\left(K_{\mathbf{yy}}(K_{\mathbf{xx}} + \lambda I)^{-1}K_{\mathbf{yy}}R_{\mathbf{yy}}^{-1}\right)
\end{aligned}
$$

and

$$
\begin{aligned}
-\frac{1}{2}\left(\log|(K_{\mathbf{xx}} + \lambda I) \otimes R_{\mathbf{yy}}|\right) &= -\frac{1}{2}\log\left(|(K_{\mathbf{xx}} + \lambda I)|^N |R|^N\right) \\
&= -\frac{N}{2}\left(\log|K_{\mathbf{xx}} + \lambda I| + \log|R|\right)
\end{aligned}
$$

where we used the fact that determinant of Kronecker product of two $N \times N$ matrices $A, B$ is: $|A \otimes B| = |A|^N |B|^N$.

Therefore the log likelihood can be expressed as

$$-\frac{N}{2}\left(\log|K_{\mathbf{xx}} + \lambda I| + \log|R|\right) - \frac{1}{2}\operatorname{tr}\left((K_{\mathbf{xx}} + \lambda I)^{-1}K_{\mathbf{yy}}R_{\mathbf{yy}}^{-1}K_{\mathbf{yy}}\right) \tag{37}$$

□

### B.4 Causal BayesCME derivations

The following proposition extend BAYESCME to the causal setting.

**Proposition C.1** (Causal BayesCME). *Denote $\mu_{gp}^{do}$ as the GP modelling $\mu_{Y|do(X)}$. Then using the $\Omega$ notations introduced in proposition 1, the posterior GP of $\mu_{gp}^{do}$ given observations $\{\mathbf{x}, \mathbf{z}, \mathbf{y}\}$ has the following mean and covariance:*

$$m_\mu^{do}((x,y)) = \Phi_{\Omega_x}(x)^\top \left( K_{\Omega_x} + \lambda I \right)^{-1} K_{\mathbf{yy}} R_{\mathbf{yy}}^{-1} r_{\mathbf{y}y} \tag{38}$$

$$\kappa_\mu^{do}((x,y),(x',y')) = \Phi_{\Omega_x}(x)^\top \Phi_{\Omega_x}(x') r_{y,y'} - \Phi_{\Omega_x}(x)^\top (K_{\Omega_x} + \lambda I)^{-1} \Phi_{\Omega_x}(x') r_{\mathbf{y}\mathbf{y}} R_{\mathbf{yy}}^{-1} r_{\mathbf{y}y'} \tag{39}$$

*In addition, the following marginal likelihood can be used for hyperparameter optimisation,*

$$-\frac{N}{2} \left( \log |K_{\Omega_x} + \lambda I| + \log |R| \right) - \frac{1}{2} \operatorname{tr} \left( (K_{\Omega_x} + \lambda I)^{-1} K_{\mathbf{yy}} R_{\mathbf{yy}}^{-1} K_{\mathbf{yy}} \right) \tag{40}$$

*Proof of Proposition C.1.* In the following we will assume $Z$ is the backdoor adjustment variable. Front-door and general cases follow analogously. Denote $\mu_{gp}((x,z),y)$ as the BAYESCME model for $\mu_{Y|X=x,Z=z}(y)$. As we have

$$\mu_{Y|do(X)=x} = \int \int \phi_y(y) p(y|x,z) p(z) dz dy \tag{41}$$

$$= \int \mu_{Y|X=x,Z=z} p(z) dz \tag{42}$$

$$= \mathbb{E}_Z[\mu_{Y|X=x,Z}] \tag{43}$$

It is thus natural to define $\mu_{gp}^{do}$ as the induced GP when we replace $\mu_{Y|X=x,Z=z}$ with $\mu_{gp}((x,z),\cdot)$,

$$\mu_{gp}^{do}(x,\cdot) = \mathbb{E}_Z[\mu_{gp}((x,Z),\cdot)] \tag{44}$$

Now we can compute the mean of $\mu_{gp}^{do}$,

$$m_\mu^{do}(x,y) = \mathbb{E}_{\mu_{gp}} \mathbb{E}_Z[\mu_{gp}(x,Z,y)] \tag{45}$$

$$= \mathbb{E}_Z \left( (k_{x\mathbf{x}} \odot k_z(Z,\mathbf{z})) (K_{\mathbf{xx}} \odot K_{\mathbf{zz}} + \lambda I)^{-1} K_{\mathbf{yy}} R_{\mathbf{yy}}^{-1} r_{\mathbf{y}y} \right) \tag{46}$$

$$= \left( (k_{x\mathbf{x}} \odot \mu_z^\top \Phi_{\mathbf{z}}) (K_{\mathbf{xx}} \odot K_{\mathbf{zz}} + \lambda I)^{-1} K_{\mathbf{yy}} R_{\mathbf{yy}}^{-1} r_{\mathbf{y}y} \right) \tag{47}$$

$$= \Phi_{\Omega_x}(x)^\top (K_{\Omega_x} + \lambda I)^{-1} K_{\mathbf{yy}} R_{\mathbf{yy}}^{-1} r_{\mathbf{y}y} \tag{48}$$

Similarly for covariance, we have,

$$\kappa_\mu^{do}((x,y),(x',y')) = \mathbb{E}_{Z,Z'}[cov(\mu_{gp}((x,Z),y), \mu_{gp}((x',Z'),y'))] \tag{49}$$

and the rest is just algebra,

$$= \Phi_{\Omega_x}(x)^\top \Phi_{\Omega_x}(x') r_{y,y'} - \Phi_{\Omega_x}(x)^\top (K_{\Omega_x} + \lambda I)^{-1} \Phi_{\Omega_x}(x') r_{\mathbf{y}\mathbf{y}} R_{\mathbf{yy}}^{-1} r_{\mathbf{y}y'} \tag{50}$$

$\square$

### B.5 BayesIME derivation

Now we have derived the Causal BAYESCME, it is time to compute $\langle f, \mu_{gp}^{do}(x,\cdot)\rangle$ where $f \in \mathcal{H}_{k_y}$. This requires us to be able to compute $\langle f, r_y(\cdot,y)\rangle$ which corresponds to the following:

$$\langle f, r_y(\cdot,y)\rangle_{\mathcal{H}_{k_y}} = \left\langle f, \int k_y(\cdot,u) k_y(u,y)\nu(du)\right\rangle \tag{51}$$

$$= \int f(u) k_y(u,y)\nu(du) \tag{52}$$

when $f$ is a KRR learnt from $\mathcal{D}_2$, i.e $f(y) = k_{y\tilde{\mathbf{y}}}(K_{\tilde{\mathbf{y}}\tilde{\mathbf{y}}} + \lambda_f I)^{-1}\mathbf{t}$, we have

$$= \mathbf{t}^\top (K_{\tilde{\mathbf{y}}\tilde{\mathbf{y}}} + \lambda_f I)^{-1} \int k_{\tilde{\mathbf{y}}u} k_y(u,y) \nu(du) \tag{53}$$

$$= \mathbf{t}^\top (K_{\tilde{\mathbf{y}}\tilde{\mathbf{y}}} + \lambda_f I)^{-1} r_{\tilde{\mathbf{y}}y} \tag{54}$$

Now we are ready to derive BAYESIME.

**Proposition 4.** *Given dataset $D_1 = \{(x_i, y_i, z_i)\}_{i=1}^N$ and $D_2 = \{(\tilde{y}_j, t_j)\}_{j=1}^M$, if $f$ is a KRR learnt from $\mathcal{D}_2$ and $\mu_{Y|do(X)}$ modelled as a V-GP using $\mathcal{D}_1$, then $g = \langle f, \mu_{Y|do(X)} \rangle \sim \mathcal{GP}(m_2, \kappa_2)$ where,*

$$m_2(x) = \Phi_{\Omega_x}(x)^\top (K_{\Omega_x} + \lambda I)^{-1} K_{\mathbf{yy}} R_{\mathbf{yy}}^{-1} R_{\mathbf{y}\tilde{\mathbf{y}}} A \tag{55}$$

$$\kappa_2(x, x') = B \Phi_{\Omega_x}(x)^\top \Phi_{\Omega_x}(x) - C \Phi_{\Omega_x}(x)^\top (K_{\Omega_x} + \lambda I)^{-1} \Phi_{\Omega_x}(x') \tag{56}$$

*where $A = (K_{\tilde{\mathbf{y}}\tilde{\mathbf{y}}} + \lambda_f I)^{-1} \mathbf{t}$, $B = A^\top R_{\tilde{\mathbf{y}}\tilde{\mathbf{y}}} A$ and $C = A^\top R_{\tilde{\mathbf{y}}\mathbf{y}} R_{\mathbf{yy}}^{-1} R_{\mathbf{y}\tilde{\mathbf{y}}} A$*

*Proof of Proposition 4.* Using the $\mu_{gp}^{do}$ notation from Proposition $C.1$, we can write the inner product as $\langle \mu_{gp}^{do}(x, \cdot), f \rangle$, where the mean is,

$$m_2(x) = \mathbb{E}[\mu_{gp}^{do}(x, \cdot)]^\top f \tag{57}$$

$$= \Phi_{\Omega_x}(x)^\top (K_{\Omega_x} + \lambda I)^{-1} K_{\mathbf{yy}} R_{\mathbf{yy}}^{-1} R(\mathbf{y}, \cdot)^\top f \tag{58}$$

$$= \Phi_{\Omega_x}(x)^\top (K_{\Omega_x} + \lambda I)^{-1} K_{\mathbf{yy}} R_{\mathbf{yy}}^{-1} R_{\mathbf{y}\tilde{\mathbf{y}}} (K_{\tilde{\mathbf{y}}\tilde{\mathbf{y}}} + \lambda_f I)^{-1} \mathbf{t} \tag{59}$$

where we used the fact $f$ is a KRR learnt from $\mathcal{D}_2$. The covariance can then be computed by realising $cov(f^\top \mu_{gp}^{do}(x, \cdot), f^\top \mu_{gp}^{do}(x', \cdot)) = f^\top cov(\mu_{gp}^{do}(x, \cdot), \mu_{gp}^{do}(x', \cdot)) f$. $\qquad \square$

## B.6 BayesIMP Derivations

BAYESIMP can be understood as a model characterising the RKHS inner product of Gaussian Processes. In the following, we will first introduce some general theory of inner product of GPs, and introduce a finite dimensional scheme later on. Finally, we will show how BAYESIMP can be derived right away from this general framework.

Before that, we will showcase the following identity for computing variance of inner products of independent multivariate Gaussians,

**Proposition C.2.** *Let $\mu_X := \mathbb{E}[X]$ and $\Sigma_X := Var(X)$ be the mean and variance of a multivariate Gaussian rv, similarly $\mu_Y, \Sigma_Y$ for Gaussian rv $Y$. If $X$ and $Y$ are independent, then the variance of their inner product is given by the following expression,*

$$Var(X^\top Y) = \mu_X^\top \Sigma_Y \mu_X + \mu_Y^\top \Sigma_X \mu_Y + tr\left(\Sigma_Y \Sigma_X\right) \tag{60}$$

*Moreover, the covariance between $X^\top Y_1$, $X^\top Y_2$ follows a similar form,*

$$cov(X^\top Y_1, X^\top Y_2) = \mu_X^\top \Sigma_{Y_1 Y_2} \mu_X + \mu_{Y_1}^\top \Sigma_X \mu_{Y_2} + tr(\Sigma_X \Sigma_{Y_1 Y_2}) \tag{61}$$

*Proof.*

$$\begin{aligned}
\text{Var}\left[X^\top Y\right] &= \mathbb{E}\left[\left(X^\top Y\right)^2\right] - \mathbb{E}\left[X^\top Y\right]^2 \\
&= \mathbb{E}\left[X^\top Y Y^\top X\right] - \left(\mathbb{E}[X]^\top \mathbb{E}[Y]\right)^2 \\
&= \mathbb{E}\left[\text{tr}\left(X X^\top Y Y^\top\right)\right] - \left(\mu_X^\top \mu_Y\right)^2 \\
&= \text{tr}\left(\mathbb{E}\left[X X^\top\right] \mathbb{E}\left[Y Y^\top\right]\right) - \left(\mu_X^\top \mu_Y\right)^2 \\
&= \text{tr}\left(\left(\mu_X \mu_X^\top + \Sigma_X\right)\left(\mu_Y \mu_Y^\top + \Sigma_Y\right)\right) - \left(\mu_X^\top \mu_Y\right)^2 \\
&= \text{tr}\left(\mu_X \mu_X^\top \mu_Y \mu_Y^\top\right) + \text{tr}\left(\mu_X \mu_X^\top \Sigma_Y\right) + \text{tr}\left(\Sigma_X \mu_Y \mu_Y^\top\right) + \text{tr}\left(\Sigma_X \Sigma_Y\right) - \left(\mu_X^\top \mu_Y\right)^2 \\
&= \left(\mu_X^\top \mu_Y\right)^2 + \text{tr}\left(\mu_X^\top \Sigma_Y \mu_X\right) + \text{tr}\left(\mu_Y^\top \Sigma_X \mu_Y\right) + \text{tr}\left(\Sigma_X \Sigma_Y\right) - \left(\mu_X^\top \mu_Y\right)^2 \\
&= \mu_X^\top \Sigma_Y \mu_X + \mu_Y^\top \Sigma_X \mu_Y + \text{tr}\left(\Sigma_X \Sigma_Y\right)
\end{aligned}$$

$$\tag{62}$$

Generalising to the case for covariance is straight forward. $\qquad \square$

**RKHS inner product of Gaussian Processes**

Let $f_1 \sim GP(m_1, \kappa_1)$ and $f_2 \sim GP(m_2, \kappa_2)$. We assume that $f$ and $g$ are both supported within the RKHS $\mathcal{H}_k$. Can we characterise the distribution of $\langle f_1, f_2 \rangle_{\mathcal{H}_k}$?

This situation would arise if $f_1$ and $f_2$ arise as GP posteriors in a regression model corresponding to the priors $f_1 \sim GP(0, r_1)$, $f_2 \sim GP(0, r_2)$ where $r_1, r_2$ satisfy the nuclear dominance property. In particular, we could choose

$$
\begin{aligned}
r_1(u, v) &= \int k(u, z) k(z, v) \nu_1(dz), \\
r_2(u, v) &= \int k(u, z) k(z, v) \nu_2(dz).
\end{aligned}
$$

Posterior means in that case can be expanded as

$$
m_1 = \sum \alpha_i r_1(\cdot, x_i), \qquad m_2 = \sum \beta_j r_2(\cdot, y_j).
$$

We assume that $f_1$ and $f_2$ are independent, i.e. they correspond to posteriors computed on independent data. Then

$$
\begin{aligned}
\mathbb{E} \langle f_1, f_2 \rangle_{\mathcal{H}_k} &= \langle m_1, m_2 \rangle_{\mathcal{H}_k} \\
&= \left\langle \sum \alpha_i r_1(\cdot, x_i), \sum \beta_j r_2(\cdot, y_j) \right\rangle_{\mathcal{H}_k} \\
&= \alpha^\top Q \beta,
\end{aligned}
$$

where

$$
\begin{aligned}
Q_{ij} = q(x_i, y_j) &:= \langle r_1(\cdot, x_i), r_2(\cdot, y_j) \rangle_{\mathcal{H}_k} \\
&= \left\langle \int k(\cdot, z) k(z, x_i) \nu_1(dz), \int k(\cdot, z') k(z', y_j) \nu_2(dz') \right\rangle_{\mathcal{H}_k} \\
&= \int \int \langle k(\cdot, z), k(\cdot, z') \rangle_{\mathcal{H}_k} k(z, x_i) k(z', y_j) \nu_1(dz) \nu_2(dz') \\
&= \int \int k(z, z') k(z, x_i) k(z', y_j) \nu_1(dz) \nu_2(dz').
\end{aligned}
$$

The variance would be given, in analogy to the finite dimensional case, by

$$
\mathrm{var} \langle f_1, f_2 \rangle_{\mathcal{H}_k} = \langle m_1, \Sigma_2 m_1 \rangle_{\mathcal{H}_k} + \langle m_2, \Sigma_1 m_2 \rangle_{\mathcal{H}_k} + \mathrm{tr}(\Sigma_1 \Sigma_2),
$$

with $\Sigma_1 f = \int \kappa_1(\cdot, u) f(u) du$ and similarly for $\Sigma_2$. Thus

$$
\begin{aligned}
\langle m_1, \Sigma_2 m_1 \rangle_{\mathcal{H}_k} &= \left\langle \sum \alpha_i r_1(\cdot, x_i), \sum \alpha_j \int \kappa_2(\cdot, u) r_1(u, x_j) du \right\rangle_{\mathcal{H}_k} \\
&= \sum \sum \alpha_i \alpha_j \int \langle r_1(\cdot, x_i), \kappa_2(\cdot, u) \rangle_{\mathcal{H}_k} r_1(u, x_j) du.
\end{aligned}
$$

Now, given that kernel $\kappa_2$ depends on $r_2$ in a simple way, it should be possible to write down the full expression similarly as for $Q_{ij}$ above. In particular

$$
\kappa_2(\cdot, u) = r_2(\cdot, u) - r_2(\cdot, \mathbf{y}) \left( R_{2, \mathbf{yy}} + \sigma_2^2 I \right)^{-1} r_2(\mathbf{y}, u).
$$

Hence

$$
\langle r_1(\cdot, x_i), \kappa_2(\cdot, u) \rangle_{\mathcal{H}_k} = q(x_i, u) - q(x_i, \mathbf{y}) \left( R_{2, \mathbf{yy}} + \sigma_2^2 I \right)^{-1} r_2(\mathbf{y}, u).
$$

However, this further requires approximating integrals of the type

$$
\int q(x_i, u) r_1(u, x_j) du = \int \int \int \int k(z, z') k(z, x_i) k(z', u) k(u, z'') k(z'', x_j) \nu_1(dz) \nu_2(dz') \nu_1(dz'') du,
$$

etc. Thus, while possible in principle, this approach to compute the variance is cumbersome.

## A finite dimensional approximation

To approximate the variance, hence, it is simpler to consider finite-dimensional approximations to $f_1$ and $f_2$. Namely, collate $\{x_i\}$ and $\{y_j\}$ into a single set of points $\xi$ (note that we could here take an arbitrary set of points), and consider finite-dimensional GPs given by

$$\tilde{f}_1 = \sum a_j k\left(\cdot, \xi_j\right), \quad \tilde{f}_2 = \sum b_j k\left(\cdot, \xi_j\right),$$

where we selects distribution of $a$ and $b$ such that evaluations of $\tilde{f}_1$ and $\tilde{f}_2$ on $\xi$, $K_{\xi\xi}a$ and $K_{\xi\xi}b$ respectively, have the same distributions as evaluations of $f_1$ and $f_2$ on $\xi$. In particular, we take

$$a \sim \mathcal{N}\left(K_{\xi\xi}^{-1}m_1\left(\xi\right), K_{\xi\xi}^{-1}\mathcal{K}_{1,\xi\xi}K_{\xi\xi}^{-1}\right), \quad b \sim \mathcal{N}\left(K_{\xi\xi}^{-1}m_2\left(\xi\right), K_{\xi\xi}^{-1}\mathcal{K}_{2,\xi\xi}K_{\xi\xi}^{-1}\right),$$

where we denoted by $m_1\left(\xi\right)$ a vector such that $\left[m_1\left(\xi\right)\right]_i = m_1\left(\xi_i\right)$ and by $\mathcal{K}_{1,\xi\xi}$ a matrix such that $\left[\mathcal{K}_{1,\xi\xi}\right]_{ij} = \kappa_1\left(\xi_i, \xi_j\right)$.

Then, clearly

$$\begin{aligned}
\left\langle \tilde{f}_1, \tilde{f}_2 \right\rangle_{\mathcal{H}_k} &= a^\top K_{\xi\xi}b \\
&= \left(K_{\xi\xi}^{1/2}a\right)^\top \left(K_{\xi\xi}^{1/2}b\right),
\end{aligned}$$

and now we are left with the problem of computing the mean and the variance of inner product between two independent Gaussian vectors, as given in Proposition $C.2$. We have

$$\begin{aligned}
\mathbb{E}\left\langle \tilde{f}_1, \tilde{f}_2 \right\rangle_{\mathcal{H}_k} &= \left(K_{\xi\xi}^{1/2}K_{\xi\xi}^{-1}m_1\left(\xi\right)\right)^\top \left(K_{\xi\xi}^{1/2}K_{\xi\xi}^{-1}m_2\left(\xi\right)\right) \\
&= m_1\left(\xi\right)^\top K_{\xi\xi}^{-1}K_{\xi\xi}K_{\xi\xi}^{-1}m_2\left(\xi\right) \\
&= m_1\left(\xi\right)^\top K_{\xi\xi}^{-1}m_2\left(\xi\right),
\end{aligned}$$

and

$$\begin{aligned}
\mathrm{var}\left\langle \tilde{f}_1, \tilde{f}_2 \right\rangle_{\mathcal{H}_k} &= \left(K_{\xi\xi}^{1/2}K_{\xi\xi}^{-1}m_1\left(\xi\right)\right)^\top K_{\xi\xi}^{-1/2}\mathcal{K}_{2,\xi\xi}K_{\xi\xi}^{-1/2}\left(K_{\xi\xi}^{1/2}K_{\xi\xi}^{-1}m_1\left(\xi\right)\right) \\
&+ \left(K_{\xi\xi}^{1/2}K_{\xi\xi}^{-1}m_2\left(\xi\right)\right)^\top K_{\xi\xi}^{-1/2}\mathcal{K}_{1,\xi\xi}K_{\xi\xi}^{-1/2}\left(K_{\xi\xi}^{1/2}K_{\xi\xi}^{-1}m_2\left(\xi\right)\right) \\
&+ \mathrm{tr}\left(K_{\xi\xi}^{-1/2}\mathcal{K}_{1,\xi\xi}K_{\xi\xi}^{-1/2}K_{\xi\xi}^{-1/2}\mathcal{K}_{2,\xi\xi}K_{\xi\xi}^{-1/2}\right) \\
&= m_1\left(\xi\right)^\top K_{\xi\xi}^{-1}\mathcal{K}_{2,\xi\xi}K_{\xi\xi}^{-1}m_1\left(\xi\right) \\
&+ m_2\left(\xi\right)^\top K_{\xi\xi}^{-1}\mathcal{K}_{1,\xi\xi}K_{\xi\xi}^{-1}m_2\left(\xi\right) \\
&+ \mathrm{tr}\left(\mathcal{K}_{1,\xi\xi}K_{\xi\xi}^{-1}\mathcal{K}_{2,\xi\xi}K_{\xi\xi}^{-1}\right).
\end{aligned}$$

## Coming back to BayesIMP

Now coming back to the derivation of BayesIMP. We will first provide two finite approximation of $f$ and $\mu_{gp}^{do}(x, \cdot)$ in the following two propositions. Recall these finite approximations are set up such that they match the distributions of evaluations of $f$ and $\mu_{gp}^{do}$ at $\hat{\mathbf{y}} = [\mathbf{y}^\top \ \tilde{\mathbf{y}}^\top]^\top$. The latter thus act as landmark points for the finite dimensional approximations.

**Proposition C.3** (Finite dimensional approximation of $f$). *Let $\hat{\mathbf{y}} = [\mathbf{y}^\top \ \tilde{\mathbf{y}}^\top]^\top$ be the concatenation of $\mathbf{y}$ and $\tilde{\mathbf{y}}$. We can approximate $f$ with ,*

$$\tilde{f}|\mathbf{t} \sim N(m_{\tilde{f}}, \Sigma_{\tilde{f}}) \tag{63}$$

*where,*

$$m_{\tilde{f}} = \Phi_{\hat{\mathbf{y}}}K_{\hat{\mathbf{y}}\hat{\mathbf{y}}}^{-1}R_{\hat{\mathbf{y}}\tilde{\mathbf{y}}}(R_{\tilde{\mathbf{y}}\tilde{\mathbf{y}}} + \lambda_f I)^{-1}\mathbf{t} \tag{64}$$

$$\Sigma_{\tilde{f}} = \Phi_{\hat{\mathbf{y}}}K_{\hat{\mathbf{y}}\hat{\mathbf{y}}}^{-1}\bar{R}_{\hat{\mathbf{y}}\hat{\mathbf{y}}}K_{\hat{\mathbf{y}}\hat{\mathbf{y}}}^{-1}\Phi_{\hat{\mathbf{y}}}^\top \tag{65}$$

*and $\bar{R}_{\hat{\mathbf{y}}\hat{\mathbf{y}}} = R_{\hat{\mathbf{y}}\hat{\mathbf{y}}} - R_{\hat{\mathbf{y}}\tilde{\mathbf{y}}}(R_{\tilde{\mathbf{y}}\tilde{\mathbf{y}}} + \lambda_f I)^{-1}R_{\tilde{\mathbf{y}}\hat{\mathbf{y}}}.*

Similarly for $\mu_{gp}^{do}(x, \cdot)$, we have the following

**Proposition C.4** (Finite dimensional approximation of $\mu_{gp}^{do}(x, \cdot)$). *Let $\hat{\mathbf{y}} = [\mathbf{y}^\top \ \tilde{\mathbf{y}}^\top]^\top$ be the concatenation of $\mathbf{y}$ and $\tilde{\mathbf{y}}$. We can approximate $\mu_{gp}^{do}(x, \cdot)$ with ,*

$$\tilde{\mu}_{gp}^{do}(x, \cdot)| \operatorname{vec}(K_{\mathbf{yy}}) \sim N(m_{\tilde{\mu}}, \Sigma_{\tilde{\mu}}) \tag{66}$$

*where,*

$$m_{\tilde{\mu}} = \Phi_{\hat{\mathbf{y}}} K_{\hat{\mathbf{y}}\hat{\mathbf{y}}}^{-1} R_{\hat{\mathbf{y}}\mathbf{y}} R_{\mathbf{yy}}^{-1} K_{\mathbf{yy}} (K_{\Omega_x} + \lambda I)^{-1} \Phi_{\Omega_x}(x) \tag{67}$$

$$\Sigma_{\tilde{\mu}} = \Phi_{\hat{\mathbf{y}}} K_{\hat{\mathbf{y}}\hat{\mathbf{y}}}^{-1} K_{\hat{\mathbf{y}}\hat{\mathbf{y}}}^{\mu} K_{\hat{\mathbf{y}}\hat{\mathbf{y}}}^{-1} \Phi_{\hat{\mathbf{y}}}^\top \tag{68}$$

*where $K_{\hat{\mathbf{y}}\hat{\mathbf{y}}}^{\mu} = \Phi_{\Omega_x}(x)^\top \Phi_{\Omega_x}(x) R_{\hat{\mathbf{y}}\hat{\mathbf{y}}} - \left( \Phi_{\Omega_x}(x)^\top (K_{\Omega_x} + \lambda I)^{-1} \Phi_{\Omega_x}(x) \right) R_{\hat{\mathbf{y}}\mathbf{y}} R_{\mathbf{yy}}^{-1} R_{\mathbf{y}\hat{\mathbf{y}}}$*

Now we have everything we need to derive the main algorithm in our paper, the BAYESIMP. Note that we did not introduce the $\mu_{gp}^{do}$ notation in the main text to avoid confusion as we did not have space to properly define $\mu_{gp}^{do}$.

**Proposition 5** (**BAYESIMP**). *Let $f$ and $\mu_{Y|do(X)}$ be GPs learnt as above. Denote $\tilde{f}$ and $\tilde{\mu}_{Y|do(X)}$ as the finite dimensional approximation of $f$ and $\mu_{Y|do(X)}$ respectively. Then $\tilde{g} = \langle \tilde{f}, \tilde{\mu}_{Y|do(X)} \rangle$ has the following mean and covariance:*

$$m_3(x) = E_x K_{\mathbf{y}\hat{\mathbf{y}}} K_{\hat{\mathbf{y}}\hat{\mathbf{y}}}^{-1} R_{\hat{\mathbf{y}}\tilde{\mathbf{y}}} (R_{\tilde{\mathbf{y}}\tilde{\mathbf{y}}} + \lambda_f I)^{-1} \mathbf{t} \tag{69}$$

$$\kappa_3(x, x') = \underbrace{E_x \Theta_1^\top \tilde{R}_{\hat{\mathbf{y}}\hat{\mathbf{y}}} \Theta_1 E_{x'}^\top}_{\text{Uncertainty from } \mathcal{D}_1} + \underbrace{\Theta_2^{(a)} F_{xx'} - \Theta_2^{(b)} G_{xx'}}_{\text{Uncertainty from } \mathcal{D}_2} + \underbrace{\Theta_3^{(a)} F_{xx'} - \Theta_3^{(b)} G_{xx'}}_{\text{Uncertainty from Interaction}} \tag{70}$$

*where $E_x = \Phi_{\Omega_x}(x)^\top (K_{\Omega_x} + \lambda I)^{-1}, F_{xx'} = \Phi_{\Omega_x}(x)^\top \Phi_{\Omega_x}(x'), G_{xx'} = \Phi_{\Omega_x}(x)^\top (K_{\Omega_x} + \lambda I)^{-1} \Phi_{\Omega_x}(x')$, and $\Theta_1 = K_{\hat{\mathbf{y}}\hat{\mathbf{y}}}^{-1} R_{\hat{\mathbf{y}}\mathbf{y}} R_{\mathbf{yy}}^{-1} K_{\mathbf{yy}}, \Theta_2^{(a)} = \Theta_4^\top R_{\hat{\mathbf{y}}\hat{\mathbf{y}}} \Theta_4, \Theta_2^{(b)} = \Theta_4^\top R_{\hat{\mathbf{y}}\mathbf{y}} R_{\mathbf{yy}}^{-1} R_{\mathbf{y}\hat{\mathbf{y}}} \Theta_4$ and $\Theta_3^{(a)} = tr(K_{\hat{\mathbf{y}}\hat{\mathbf{y}}}^{-1} R_{\hat{\mathbf{y}}\hat{\mathbf{y}}} K_{\hat{\mathbf{y}}\hat{\mathbf{y}}}^{-1} \bar{R}_{\hat{\mathbf{y}}\hat{\mathbf{y}}}), \Theta_3^{(b)} = tr(R_{\hat{\mathbf{y}}\mathbf{y}} R_{\mathbf{yy}}^{-1} R_{\mathbf{y}\hat{\mathbf{y}}} K_{\hat{\mathbf{y}}\hat{\mathbf{y}}}^{-1} \bar{R}_{\hat{\mathbf{y}}\hat{\mathbf{y}}} K_{\hat{\mathbf{y}}\hat{\mathbf{y}}}^{-1})$ and $\Theta_4 = K_{\hat{\mathbf{y}}\hat{\mathbf{y}}}^{-1} R_{\hat{\mathbf{y}}\tilde{\mathbf{y}}} (K_{\tilde{\mathbf{y}}\tilde{\mathbf{y}}} + \lambda_f)^{-1} \mathbf{t}$. $\bar{R}_{\hat{\mathbf{y}}\hat{\mathbf{y}}}$ is the posterior covariance of $f$ evaluated at $\hat{\mathbf{y}}$*

*Proof of Proposition 5.* Since $\tilde{g} = \langle \tilde{f}, \tilde{\mu}_{gp}^{do} \rangle$ is an inner product between two finite dimensional GPs, we know the variance (as given by Proposition $C.2$) is characterised by,

$$var(g) = m_{\tilde{\mu}}^\top \Sigma_{\tilde{f}} m_{\tilde{\mu}} + m_{\tilde{f}}^\top \Sigma_{\tilde{\mu}} m_{\tilde{f}} + \operatorname{tr}(\Sigma_{\tilde{f}} \Sigma_{\tilde{\mu}}) \tag{71}$$

Expanding out each terms we get Proposition 5:

$$m_{\tilde{\mu}}^\top \Sigma_{\tilde{f}} m_{\tilde{\mu}} = E_x \Theta_1^\top \tilde{R}_{\hat{\mathbf{y}}\hat{\mathbf{y}}} \Theta_1 E_{x'}^\top \tag{72}$$

$$m_{\tilde{f}}^\top \Sigma_{\tilde{\mu}} m_{\tilde{f}} = \Theta_2^{(a)} F_{xx'} - \Theta_2^{(b)} G_{xx'} \tag{73}$$

$$\tag{74}$$

while the first two terms resembles the uncertainty obtained from IMP and BAYESIME, the trace term is new and we will expand it out here,

$$\operatorname{tr}(\Sigma_{\tilde{f}} \Sigma_{\tilde{\mu}}) = \operatorname{tr}\left( \Phi_{\hat{\mathbf{y}}} K_{\hat{\mathbf{y}}\hat{\mathbf{y}}}^{-1} K_{\hat{\mathbf{y}}\hat{\mathbf{y}}}^{\mu} K_{\hat{\mathbf{y}}\hat{\mathbf{y}}}^{-1} \Phi_{\hat{\mathbf{y}}}^\top \Phi_{\hat{\mathbf{y}}} K_{\hat{\mathbf{y}}\hat{\mathbf{y}}}^{-1} \bar{R}_{\hat{\mathbf{y}}\hat{\mathbf{y}}} K_{\hat{\mathbf{y}}\hat{\mathbf{y}}}^{-1} \Phi_{\hat{\mathbf{y}}}^\top \right) \tag{75}$$

$$= \operatorname{tr}\left( K_{\hat{\mathbf{y}}\hat{\mathbf{y}}}^{-1} K_{\hat{\mathbf{y}}\hat{\mathbf{y}}}^{\mu} K_{\hat{\mathbf{y}}\hat{\mathbf{y}}}^{-1} \bar{R}_{\hat{\mathbf{y}}\hat{\mathbf{y}}} \right) \tag{76}$$

$$= \operatorname{tr}\left( K_{\hat{\mathbf{y}}\hat{\mathbf{y}}}^{-1} \left( F_{xx'} R_{\hat{\mathbf{y}}\hat{\mathbf{y}}} - G_{xx'} R_{\hat{\mathbf{y}}\mathbf{y}} R_{\mathbf{yy}}^{-1} R_{\mathbf{y}\hat{\mathbf{y}}} \right) K_{\hat{\mathbf{y}}\hat{\mathbf{y}}}^{-1} \bar{R}_{\hat{\mathbf{y}}\hat{\mathbf{y}}} \right) \tag{77}$$

$$= \Theta_3^{(a)} F_{xx'} - \Theta_3^{(b)} G_{xx'} \tag{78}$$

$\square$

# C  Details on Experimental setup

## C.1  Details on Ablation Study

### C.1.1  Data Generating Process

We use the following causal graphs, $X \rightarrow Y$ and $Y \rightarrow T$, to demonstrate a simple scenario for our data fusion setting. As linking functions, we used for $\mathcal{D}_1$, $Y = xcos(\pi x) + \epsilon_1$ and for $\mathcal{D}_2$, $T = 0.5 * y * cos(y) + \epsilon_2$. where $\epsilon_i \sim \mathcal{N}(0, \sigma_i)$. Here below we plotted the data for illustration purposes.

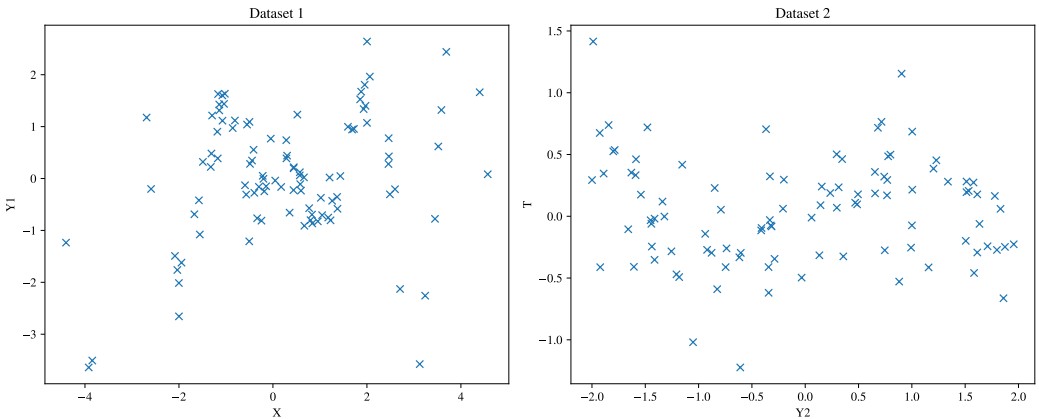

Figure 9: (Left) Illustration of $\mathcal{D}_1$ (Right) Illustration of $\mathcal{D}_2$

### C.1.2  Explanation on the extrapolation effect

In the main text we referred to the case where IMP is better than BAYESIME as **extrapolation effect**. We note from the figure above that in $\mathcal{D}_1$ we have $x$ around $-4$ being mapped onto $y$ values around $-3$. Note however, that in $\mathcal{D}_2$, we do not observe any values $\tilde{Y}$ below $-2$. Hence, because IMP uses a GP model for $\mathcal{D}_2$ we are able to account for this mismatch in support and hence attribute more uncertainty to this region, i.e. we see the spike in uncertainty in Fig.5 for IMP.

### C.1.3  Calibration Plots

To investigate the accuracy of the uncertainty quantification in the proposed methods, we perform a (frequentist) calibration analysis of the credible intervals stemming from each method. Fig. 10 gives the calibration plots of the Sampling methods (sampling-based method of [15]) as well as the three proposed methods. On the x-axis is the portion of the posterior mass, corresponding to the width of the credible interval. We will interpret that as a nominal coverage probability of the true function values. On the y-axis is the true coverage probability estimated using the percentage of the times true function values do lie within the corresponding credible intervals. A perfectly calibrated method should have nominal coverage probability equal to the true coverage probability, i.e. being closer to the diagonal line is better.

## C.2  Details on Synthetic Data experiments

### C.2.1  Data Generating Process for simple synthetic dataset

For the first simple synthetic dataset (See Fig.6 (Top)) we used the following data generating graph is defined as.

- $X \rightarrow U : U = 2 * X + \epsilon$
- $Z \rightarrow X : X = 3 * \cos(Z) + \epsilon$
- $\{Z, U\} \rightarrow Y : Y = U + \exp(-Z) + \epsilon$

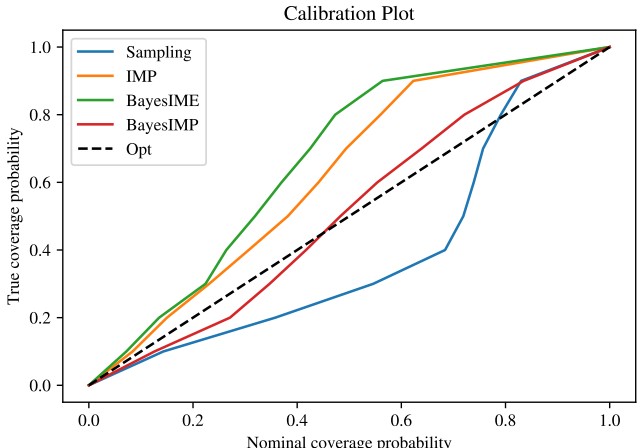

Figure 10: Calibration plots of Sampling method as well as our 3 proposed methods. We clearly see that BAYESIMP is the best calibrated method amongst all other methods.

- $Y \to T : T = \cos(Y) - \exp(-y/20) + \epsilon$

where $\epsilon \sim \mathcal{N}(0, \sigma^2)$ and $Z \sim \mathcal{U}[-4, 4]$, where for $\mathcal{D}_2$ we have that $\tilde{Y} \sim \mathcal{U}[-10, 10]$. In addition, with probability $\pi = 1/2$ we shift $U$ by $+1$ horizontally and $-3$ vertically to thus create the multimodality in the data. In order to generate from the interventional distribution, we simply remove the edge from $Z \to X$ and fix the value of $x$.

### C.2.2 Data Generating Process for harder synthetic dataset from [15]

For the first simple synthetic dataset (Fig.6(Bottom)) we used the same data generating format as in [15].

- $U_1 = \epsilon_1$
- $U_2 = \epsilon_2$
- $F = \epsilon_3$
- $A = F^2 + U_1 + \epsilon_A$
- $B = U_2 + \epsilon_B$
- $C = \exp(-B) + \epsilon_C$
- $D = \exp(-C)/10 + \epsilon_D$
- $E = \cos(A) + C/10\epsilon_E$
- $Y_1 = \cos(D) + \sin(E) + U_1 + U_2$
- $Y_2 = \cos(D) + \sin(E) + U_1 + U_2 + 2\pi$
- $T = 6 * \sin(3 * Y) + \epsilon$

where the noise is fixed to be $\mathcal{N}(0, 1)$ and where we switch with $\pi = 1/2$ from mode $Y_1$ and $Y_2$, where $\tilde{Y} \sim \mathcal{U}[-2, 9]$ for $\mathcal{D}_2$.

### C.3 Details on Healthcare Data experiments

### C.3.1 Data Generating Process

For the healthcare dataset, $\mathcal{D}_1$, (Fig.1) we used the same data generating format as in [15] with the difference that we make *statin* continuous and increased the age range.

- $age = \mathcal{U}[15, 75]$

- $bmi = \mathcal{N}(27 - 0.01 * age, 0.7)$
- $aspirin = \sigma(-8.0 + 0.1 * age + 0.03 * bmi)$
- $statin = -13 + 0.1 * age + 0.2 * bmi$
- $cancer = \sigma(2.2 - 0.05 * age + 0.01 * bmi - 0.04 * statin + 0.02 * aspirin)$
- $PSA = \mathcal{N}(6.8 + 0.04 * age - 0.15 * bmi - 0.6 * statin + 0.55 * aspirin + cancer, 0.4)$

As for the second dataset, $\mathcal{D}_2$ we firstly fit a GP on the data collected from [7]. Once we have the posterior GP, we can then use it as a generator for the $\mathcal{D}_2$ as it takes as input $PSA$. This generator hence acts as a link between $\mathcal{D}_1$ and $\mathcal{D}_2$. This way we are able to create a simulator that allows us to obtain samples from $\mathbb{E}[Cancer\ volume|do(Statin)]$ for our causal BO setup.

### C.4 Bayesian Optimisation experiments with IMP and BAYESIME

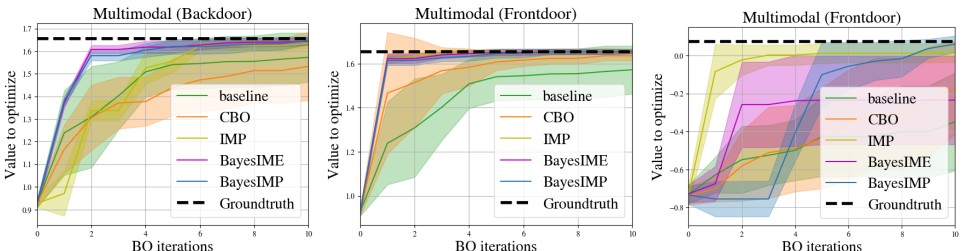

Figure 11: (Left) Simple graph using backdoor adjustment (Middle) Simple graph using front-door adjustment (Right) Harder graph using front-door adjustment. BAYESIMP strikes the right balance between IMP and BAYESIME and all three perform better than CBO and the GP baseline.

The main text compares BAYESIMP to CBO and the baseline GP with no learnt prior in the Bayesian Optimisation experiments. Here, we include IMP and BAYESIME (i.e. simplified versions of BAYESIMP that account for only one source of uncertainty each) in those comparisons. We see from Fig.11 that BAYESIMP is comparable to IMP and BAYESIME in most cases. While BAYESIMP is not the best performing method in every scenario, it does hit a good middle ground between the first two proposed methods. For Fig.11 (Left, Middle) we used $N = 100$ and $M = 50$. In the left figure, BAYESIME and BAYESIMP are very similar, whereas IMP is considerably worst. In the middle figure, all methods seems to perform well without much difference. In the right figure, we have $N = 500$ and $M = 50$ and this is a case where IMP is best, while BAYESIME appears to get stuck in a local optimum (recall that BAYESIME does not take into account uncertainty in $\mathcal{D}_2$ where there is little data). We note that all three methods converge faster than the current SOTA CBO.