# OpenReview forum: "BayesIMP: Uncertainty Quantification for Causal Data Fusion"
_NeurIPS.cc/2021/Conference — NeurIPS 2021 Poster_

### Official Review · Reviewer_7uEU · 2021-06-24

**Rating:** 6
**Confidence:** 3

**Summary:**

The paper proposes a Bayesian approach for estimating the treatment effect by combining two data sources. The main advantage of the Bayesian approach over the frequentist counterpart [17] is its ability to quantify uncertainty. The main machinery was GP which replaces KRR used in existing works.

**Limitations And Societal Impact:**

The limitations are adequately discussed and the potential negative societal impact is briefly discussed.

**Main Review:**

The paper is very well written, experiments are adequate to illustrate the strength of the approaches, and the proposed idea has some interesting contributions (i.e., causal uncertainty quantification in the data fusion context) to the existing literature. In terms of potential applications of the proposed methods, I believe there are plenty in medicine and other fields where data are collected from multiple sources.  Therefore, I believe overall the quality, clarity, and significance of this work are high.

One concern is, however, regarding the originality. The novel elements of this paper are all based on established techniques, e.g., the IMP extends [24] and IME extends [25-27]. The idea of using RKHS is also not particularly new. However, the utilization of these various existing techniques in a meaningful way in this particular context is, in my opinion, novel.

======update=====
Thanks for the authors' responses. I keep my original score.

**Time Spent Reviewing:**

5

---

> ### Author Response · Authors · 2021-08-10
> **Response to reviewer 7uEU**
>
>
> > “One concern is, however, regarding the originality. The novel elements of this paper are all based on established techniques, e.g., the IMP extends [24] and IME extends [25-27]. The idea of using RKHS is also not particularly new. However, the utilization of these various existing techniques in a meaningful way in this particular context is, in my opinion, novel.”
>
>
> We would like to thank the reviewer for commending our novel and meaningful use of various RKHS techniques in the setting of causal data fusion. However, we respectfully disagree that BayesIME, or more precisely, BayesCME, is just an extension of [25-27] for the following reasons:
>
>
> - Regarding [26], while they also model mean embeddings using priors in the RKHSs, their focus is on standard mean embeddings, and conditional ones are not considered. In addition, their approach can be viewed as regression from x_i to the _evaluation_ of the empirical mean embedding ($\hat{\mu}(x_i) = \frac{1}{N}\sum_{j=1}^n k(x_j, x_i)$) and hence extending their approach to conditional mean embeddings would result in a different model than ours, since they do not consider an RKHS-valued ridge regression approach like in our work.
> - In [27], the authors’ focus is on devising an optimisation objective for multiclass conditional embeddings (MCE), which are conditional mean embeddings with a categorical target output space. This is very different from our work in which we did not restrict ourselves to a categorical target output space. Contrary to [27], we established a method for using priors over joint RKHSs, which in itself is challenging and required the use of the notion of nuclear dominance (an established result we applied from [25]) to ensure our method is mathematically well defined. This construction is not considered in [27]. Furthermore, the optimisation objectives of both methods are also different due to the different model setup. In [27], they considered a modified cross-entropy loss because of how MCE can be used as asymptotic probabilities while we optimise our model based on Gaussian likelihood, therefore we believe our work cannot be seen as just an extension of [27].
>
>
>
>
> > “The paper is very well written, experiments are adequate to illustrate the strength of the approaches, and the proposed idea has some interesting contributions (i.e., causal uncertainty quantification in the data fusion context) to the existing literature. In terms of potential applications of the proposed methods, I believe there are plenty in medicine and other fields where data are collected from multiple sources. Therefore, I believe overall the quality, clarity, and significance of this work are high.”
>
>
> We would like to thank the reviewer again for their time reviewing our paper and hope to have clarified their concerns. We also believe that there are plenty of applications in medical science where our method could be used to quantify uncertainty from multiple sources and hence would like to ask the reviewer to revise their score in our favour if we were able to address all their concerns.

---

### Official Review · Reviewer_JKck · 2021-07-05

**Rating:** 6
**Confidence:** 4

**Summary:**

The paper studies the problem of uncertainty quantification for causal data fusion.  They introduce Bayesian Interventional Mean Processes, a framework that represents the interventional distributions using kernel mean embeddings and probabilistic integration. The framework is able to measure the uncertainty within the causal graph in each dataset. In experiments, the author demonstrates the performance improvement of the proposed method on the causal Bayesian optimization task.

**Limitations And Societal Impact:**

Yes, they do.

**Main Review:**

The setup is clearly related to the problem of instrumental variable (IV) regression. Besides the difference that the regression at different stages are based on different datasets, the connection with IV can be seen by changing the treatment variable to an instrumental variable and changing the mediating treatment variable to a treatment variable.

If this is the case, I think the method may be better suited to be presented in the potential outcome (PO) framework rather than the framework of structural causal models. Under the PO framework, you should provide the same assumptions for instrumental variable (IV) regression. For example, the treatment variable (the instrumental variable in IV) has no direct effect on the target variable; there are no unmeasured confounders for the treatment variable (the instrumental variable in IV) and the target variable. Because the paper is about uncertainty quantification, I found putting down these formal assumptions becomes more important. For instance, the biggest uncertainty in causal inference can be due to some unmeasured confounders of the treatment variable (the instrumental variable in IV) and the target variable. Later on, I see you provide two assumptions in lines 155-157. I do believe these assumptions should be made earlier, e.g., in Section 2.1.

In the introduction, I think the author should provide more detailed explanations of epistemic uncertainties and aleatoric uncertainties. Bayesian methods are not mainstream in causal inference. I don’t think causal inference researchers are familiar with these two concepts. In line 74, you mention “[17] considers a frequentist approach, which does not account for epistemic uncertainties”.  Not all the frequentist approaches measure the uncertainty using the asymptotic confidence intervals. For example, bootstraps or subsampling can be used to quantify uncertainties. It is unclear why we should use the Bayesian approach to quantify the uncertainty instead of the frequentist approach. It is possible that the user chooses a wrong prior on the Gaussian process models, then the Bayesian uncertainty is also not very well calibrated, i.e., the epistemic uncertainties are not quantified in the right class of models.

The proposed method is presented in a clear way. However, I don’t think making something Bayesian is very novel technically, given what has been done on Interventional mean embeddings and the connection between kernel ridge regression and Gaussian process regression. Is there any technical challenge due to using multiple datasets your method overcome but I didn’t realize?

In experiments, the ablation study should report the coverage rate at a range of sample sizes, dimensions and simulated functions. It is not convincing to see just the performance of one function, dimension and sample size. Causal Bayesian optimization is an interesting application but may be too complicated. I understand fast convergence in Bayesian optimization means the method estimates the function mean and variance very well. I expect to see some simple experiments on treatment effect estimation, in which you can disentangle the gain from estimating mean and variance. Also, Bayesian optimization is not effective at every iteration; it just improves the value to optimize in some steps. Overall, I think doing experiments on treatment effect estimation directly is more transparent. For your healthcare experiments, learning dosage in practice is usually not that straightforward. Increasing the dose level also increases the toxicity. There should be an unknown safety constraint on your intervention. Bayesian optimization in your healthcare example should take this into account for the method to be used in real-world applications.

There are many recent works (see below) in causal inference that try to combine multiple datasets. They are not directly very relevant to your paper. But your paper title and abstract are too general without specifying the problem setup you focus on.

Shu Yang, Peng Ding. Combining multiple observational data sources to estimate causal effects. https://arxiv.org/abs/1801.00802

Katherine Evans, BaoLuo Sun, James Robins, Eric J. Tchetgen Tchetgen. Doubly Robust Regression Analysis for Data Fusion. https://arxiv.org/abs/1808.07309

Wenshuo Guo, Serena Wang, Peng Ding, Yixin Wang, Michael I. Jordan.
Multi-Source Causal Inference Using Control Variates. https://arxiv.org/abs/2103.16689

E Rosenman, AB Owen, M Baiocchi, H Banack. Propensity score methods for merging observational and experimental datasets. arXiv preprint arXiv:1804.07863






**Time Spent Reviewing:**

4 hours

---

> ### Author Response · Authors · 2021-08-10
> **Response to reviewer JKck**
>
> > “The setup is clearly related to the problem of instrumental variable (IV) regression. Besides the difference that the regression at different stages are based on different datasets, the connection with IV can be seen by changing the treatment variable to an instrumental variable and changing the mediating treatment variable to a treatment variable.”
>
> First of all we would like to note that even though there are similarities between our proposed method and IV regression, the goal, as well as the datasets we use, are fundamentally different. We agree with the reviewer that in both problems we can use a 2 stage procedure to infer the quantities we are interested in. However, we disagree that we could simply change “the treatment variable to an instrumental variable and [change] the mediating treatment variable to a treatment variable”, because of two main reasons:
>
> 1) IV regression requires a very specific causal graphical structure i.e. in the simplest case for illustration purposes assume we have variables Z (instrument), X(Treatment), W(unobserved confounder) and Y(Target). And given the causal graph Z->X, W->[X,Y] and X->Y (Note that Z can neither be related to the confounder nor the target variable). IV regression aims to investigate the relationship of X and Y using the instrument Z when W is unobserved. However, a crucial assumption is that Z and W have to be strictly independent. Hence “simply” renaming our variables in our problem setup will not result in a IV regression setting given that the instrument will be connected to the confounder. (We have a confounder between (our) treatment and (our) mediating variable).
>
> 2) In the case that the reviewer lays out (i.e. renaming the variables in our graph) we do not have a hidden confounder between the target and the treatment as in IV regression. In the case we describe in our paper we have a hidden confounder between the (our) treatment variable and (our) mediating variable, This is not the case in IV regression.
>
> Hence, even though both problems can be solved using a 2 stage procedure they are fundamentally different in their goals as well as data provided. We would like to still thank the reviewer for pointing out these similarities and will add these comments in the revised version in order to make the distinction clear.
>
>
>
> > “If this is the case, I think the method may be better suited to be presented in the potential outcome (PO) framework rather than the framework of structural causal models. Under the PO framework, you should provide the same assumptions for instrumental variable (IV) regression. For example, the treatment variable (the instrumental variable in IV) has no direct effect on the target variable; there are no unmeasured confounders for the treatment variable (the instrumental variable in IV) and the target variable. Because the paper is about uncertainty quantification, I found putting down these formal assumptions becomes more important. For instance, the biggest uncertainty in causal inference can be due to some unmeasured confounders of the treatment variable (the instrumental variable in IV) and the target variable. Later on, I see you provide two assumptions in lines 155-157. I do believe these assumptions should be made earlier, e.g., in Section 2.1.”
>
> We would like to note that the assumption we made in lines 155-157, i.e. all paths between the treatment variable X and target variable Y are mediated through Y, directly translates into having no direct effect between (our) treatment and (our) target variables and that there are no unmeasured confounders. However, we agree that we should put this assumption on conditional independence earlier, which we will do in our revised version.
>
> ----------------------------------------------------------------------------------------------------------------------------------------------------------------
> > “In the introduction, I think the author should provide more detailed explanations of epistemic uncertainties and aleatoric uncertainties. …. Not all the frequentist approaches measure the uncertainty using the asymptotic confidence intervals. For example, bootstraps or subsampling can be used to quantify uncertainties. “
>
> We agree with the reviewer that there are indeed frequentist approaches that are able to give us uncertainty estimates, i.e. such as subsampling or bootstrapping. However, we would like to note that these frequentist methods also have both advantages and disadvantages. On the one hand, one clear advantage that the reviewer correctly lays out is the absence of priors, which could lead to potentially unwanted biases when trying to draw conclusions from bayesian models.
>
> On the other hand, however, quantifying uncertainty in a frequentist sense using subsampling or bootstrapping, requires repeated evaluations and training of models which might be computationally costly. In addition, evaluations of uncertainty in these cases are only available pointwise (not smooth) and hence not well suited for Bayesian optimization settings as discussed in [Fig, 2/4 Ref.2].  Lastly, as already pointed out in the general comments, models that use subsampling as their source of uncertainty often tend to be biased and are not necessarily well-calibrated, see [Sec 4.1 Ref.1].
>
> In comparison, GP models (Bayesian models), provide us with tractable uncertainty functions and avoid the need to use cross-validation for hyperparameter optimization. Furthermore, we would like to direct the reviewer to Fig 10 in the Appendix where we investigate the calibration of our proposed method BayesIMP. Nonetheless, as mentioned in the general comments above, any Bayesian method, including ours, can lead to biased estimates when prior knowledge misrepresents the problem at hand -- and this point will be clarified in the revised version
>
> Hence there are no clear advantages to choose one over the other without further context. However, in this work, we are interested in solving the Causal Bayesian optimisation problem for causal data fusion, which therefore motivates the Bayesian approach. As mentioned before, the use of frequentist methods for these kinds of optimization tasks have been discussed [see Ref.2], and it has been established that Bayesian methods such as GPs are favoured in this scenario. Exploring the frequentist uncertainty quantification in the realm of data fusion and causal Bayesian optimization is possible and will be an interesting direction to pursue in the future.
>
>
>
> ----------------------------------------------------------------------------------------------------------------------------------------------------------------
>
>
> > “The proposed method is presented in a clear way. However, I don’t think making something Bayesian is very novel technically, given what has been done on Interventional mean embeddings and the connection between kernel ridge regression and Gaussian process regression. Is there any technical challenge due to using multiple datasets your method overcome but I didn’t realize?”
>
>
> We respectfully disagree that our approach of “making something Bayesian is not very novel” as coming up with the Bayesian model for conditional mean embedding requires highly non-trivial derivations. To the best of our knowledge, we are the first to introduce this vector valued-GP formulation of the CME, allowing us to capture tractable uncertainty when estimating kernel mean embeddings. Furthermore, our formulation provides an alternative hyperparameter selection method for CME based on likelihood optimisation, in contrast to the existing cross-validation based approaches in standard CMEs.
>
> In addition, we note that there are technical challenges that the reviewer might have missed, which we will clarify in the revised version. In particular, we would like to direct the reviewer to lines 213-232 where we tackle the challenge of properly defining $\langle f, \mu_{Y|do(X)} \rangle$. Given that we put a GP on $ \mu_{Y|do(X)}$, we need to ensure that the resulting samples from that GP lie in the same RKHS as $f$ in order for the inner product to be well defined (both quantities of an inner product have to lie in the same space). Hence we had to make use of the notion of nuclear dominant kernels, which we discuss in Appendix B2, in order to overcome these challenges. Furthermore, the derivations are non-trivial as can be seen in Appendix B3.
>
> It is unclear if and how our specific data fusion context, and, in particular, uncertainty quantification in this context, can be tackled efficiently with frequentist methods such as subsampling etc. In particular, for our experiments on Causal Bayesian optimization, the Bayesian framework as mentioned before seems to be the preferred option as stated in [Ref. 2].
>
>
> TO BE CONTINUED
>
> Ref.1 [Hüllermeier, Eyke, and Willem Waegeman. "Aleatoric and epistemic uncertainty in machine learning: An introduction to concepts and methods." Machine Learning 110.3 (2021): 457-506.]
>
> Ref.2 [Jenatton et al. 2017“ Bayesian Optimization with Tree-structured Dependencies”]

---

> > ### Author Response · Authors · 2021-08-10
> > **Response to reviewer JKck CONTINUED**
> >
> > > “Causal Bayesian optimization is an interesting application but may be too complicated. I understand fast convergence in Bayesian optimization means the method estimates the function mean and variance very well. I expect to see some simple experiments on treatment effect estimation, in which you can disentangle the gain from estimating mean and variance”
> >
> > We believe that Bayesian optimization (BO) is a convincing method to demonstrate the merit of our uncertainty quantification as it is an essential component for the exploitation vs exploration mechanism in BO. In addition, we would like to direct the reviewer to Figure 10 in the Appendix for a more in-depth view of our ablation study. To understand the variance, we analyse the “calibration” of our uncertainty estimates by considering _frequentist coverage properties_ of the resulting credible intervals. We note that BayesIMP’s nominal coverage probability follows closely the true coverage probability, indicating well-calibrated uncertainties (the closer to the diagonal line the better) and the absence of any undesirable bias due to priors.
> >
> > In terms of mean estimation, as seen from propositions 1, 2, 4 and 5, their mean functions are very similar, differing by the correction terms induced from the fact we needed to use nuclear dominant kernels to ensure our models are well-posed. This can also be seen in Fig.5 where we can see that the means are not differing too much.
> >
> > Hence, we believe that our ablation study disentangles the contributions of the mean and variance. However, we will clarify this in more detail in the revised version.
> >
> > ----------------------------------------------------------------------------------------------------------------------------------------------------------------
> >
> > > “For your healthcare experiments, learning dosage in practice is usually not that straightforward. Increasing the dose level also increases the toxicity. There should be an unknown safety constraint on your intervention. Bayesian optimization in your healthcare example should take this into account for the method to be used in real-world applications.”
> >
> > We agree with the reviewer that learning dosage in practice is not that straightforward and in fact, [Aglietti et al 2020] considered adding a cost function in their optimisation problem setup. However, in this paper, we focused on demonstrating the merit of our method when quantifying uncertainty. Hence, for clarity, we kept our problem setup simple. In practice, we could easily add a cost function to our objective just as in [Aglietti et al 2020]’s approach. We will add this discussion in the revised version.
> >
> >
> > ----------------------------------------------------------------------------------------------------------------------------------------------------------------
> >
> >
> > > "There are many recent works (see below) in causal inference that try to combine multiple datasets. They are not directly very relevant to your paper. But your paper title and abstract are too general without specifying the problem setup you focus on.
> >
> > >Shu Yang, Peng Ding. Combining multiple observational data sources to estimate causal effects. https://arxiv.org/abs/1801.00802
> >
> > >Katherine Evans, BaoLuo Sun, James Robins, Eric J. Tchetgen Tchetgen. Doubly Robust Regression Analysis for Data Fusion. https://arxiv.org/abs/1808.07309
> >
> > >Wenshuo Guo, Serena Wang, Peng Ding, Yixin Wang, Michael I. Jordan. Multi-Source Causal Inference Using Control Variates. https://arxiv.org/abs/2103.16689
> >
> > >E Rosenman, AB Owen, M Baiocchi, H Banack. Propensity score methods for merging observational and experimental datasets. arXiv preprint arXiv:1804.07863"
> >
> > We thank the reviewer for pointing us to relevant literature on the topic of data fusion. We will add the references to the main work accordingly, and ensure that our problem setup is clear from the outset
> >
> > Lastly, we hope to have clarified the concerns of the reviewer and would like to ask the reviewer to reconsider their score.
> >
> > References:
> > Aglietti et al. 2020 Causal Bayesian Optimization, Proceedings of the Twenty Third International Conference on Artificial Intelligence and Statistics, PMLR 108:3155-3164, 2020.

---

> > > ### Comment · Reviewer_JKck · 2021-08-16
> > > **Response to the authors of Paper6134**
> > >
> > > Thank you very much for your feedback. I increase my score to 6 after reading your reply. Overall, the paper requires extensive preliminary knowledge to understand, including causal inference, data fusion, uncertainty quantification, gaussian process and bayesian optimization. Although the paper is well-written, the authors should consider providing readers with additional help (e.g. preliminary sections in Appendix or some references) to better understand each part of the paper.

---

### Official Review · Reviewer_ekhA · 2021-07-16

**Rating:** 6
**Confidence:** 3

**Summary:**

This paper tackles the problem of computing the average causal effect given data generated by distinct (fragmented) graphs which does not contain the same set of variables. It uses kernel mean embeddings to represent interventional distributions, marginalize out the common variable Y and account for statistical uncertainties.

**Main Review:**

Firstly the problem of fusing datasets consisting of different variables  from distinct data sources and using them to answer questions is a non-trivial  and important task. So this is an important and interesting work.

My main concern here is the assumption T || do(X)|Y. Note that T and Y are variables and do() is an operator. As such it is strange to see an independence claim expressed between  a variable and an operator. Normally, we rely on rules of do-calculus to remove or add do(x) operators to factors in an estimand. In this case the variables do not  belong to the same  causal model (but to two distinct graphs) and as such it is not clear how to directly apply do-calculus.

A) By T || do(X)|Y, do you mean that in the true data generating model G that generated P(X,Y,Z,T) (which is unavailable), X is not a direct cause of T and that all causal paths from X to T are mediated by Y? If so, explicitly state this. If not, please explain this to me.

B) Wouldn’t a minor modification of your results make them applicable to the case where Z is not a common cause of X and Y but related to them in the following way: X-->Z-->Y?

C) Also in the model you have considered, why are there no confounders between Y and T? Realistically, wouldn't that be the case?


More importantly, I think the authors need to put in a bit more effort to make their results more accessible to the general audience. Even in the case of causal inference researchers, not everyone can be expected to be familiar with RKHS and hence might not be able to make use of the results presented here. It is important therefore that the authors add a sentence or two after each proposition describing it in english (without RKHS jargon).


**Time Spent Reviewing:**

---

> ### Author Response · Authors · 2021-08-10
> **Response to reviewer ekhA**
>
> > “My main concern here is the assumption T || do(X)|Y. Note that T and Y are variables and do() is an operator. As such it is strange to see an independence claim expressed between a variable and an operator. Normally, we rely on rules of do-calculus to remove or add do(x) operators to factors in an estimand. In this case the variables do not belong to the same causal model (but to two distinct graphs) and as such it is not clear how to directly apply do-calculus.
> By T || do(X)|Y, do you mean that in the true data generating model G that generated P(X,Y,Z,T) (which is unavailable), X is not a direct cause of T and that all causal paths from X to T are mediated by Y? If so, explicitly state this. If not, please explain this to me.”
>
> We thank the reviewer for pointing out our unclear notation in terms of conditional independence. And yes, when stating T|| do(X) |Y, we mean that p(t|do(x),y)=p(t|y), or, in other words, that in the true data generating model P(X, Y, Z, T), all causal paths from X to T are mediated through Y. This is indeed a very important assumption that we made in the paper and will explicitly state this in the updated version of our paper.
>
> ------------------------------------------------------------------------------------------------------------------------------------------------------------------
>
> > “Wouldn’t a minor modification of your results make them applicable to the case where Z is not a common cause of X and Y but related to them in the following way: X-->Z-->Y?”
>
> In the case where X->Z->Y (with no hidden confounders) we no longer have a need for adjustment formulas and hence p(Y|do(X)=x) = p(Y|X=x). In these cases instead of interventional mean embeddings, we use standard kernel mean embeddings and the rest follows similarly. We will clarify this in the revision.
>
> ------------------------------------------------------------------------------------------------------------------------------------------------------------------
>
> > “Also in the model you have considered, why are there no confounders between Y and T? Realistically, wouldn't that be the case?”
>
> We thank the reviewer for pointing this out. In fact, we deliberately did not consider models with confounders between Y and T for the clarity of the paper. In fact, it would be straightforward to extend our setup to such a case under the assumption that “all paths from X to T are mediated through Y”. Under that assumption, the following would still hold true:
>
> $E[T|do(X))  = \int T p(T|Y, do(X))p(Y|do(X)) dYdT = \int T p(T|Y) p(Y|do(X)) dYdT = \int E[T|Y] p(Y|do(X)) dY$
>
> Therefore to estimate E[T|do(X)], we still learn a regression from Y to T, regardless of the presence of any additional variables in the second graph, since we are interested in E[T|Y] (note that the intervention is on X, and not on Y).
>
> ------------------------------------------------------------------------------------------------------------------------------------------------------------------
>
> > “Even in the case of causal inference researchers, not everyone can be expected to be familiar with RKHS and hence might not be able to make use of the results presented here. It is important therefore that the authors add a sentence or two after each proposition describing it in english (without RKHS jargon).”
>
> Due to limited space, we had to cut down on some of the details but they will be added in the revised version. In particular, we will provide additional clarifications, intuitions and background on RKHS for each of the propositions in our finalised version.
>
> We would like to again thank the reviewer for the time reviewing our work and hope to have clarified any remaining concerns of our paper. We agree with the reviewer that this is indeed “important and interesting work” and hence would like to ask the reviewer to reconsider their score.

---

### Official Review · Reviewer_wLHr · 2021-07-16

**Rating:** 6
**Confidence:** 3

**Summary:**

The paper proposes a framework for embedding interventional distributions into RKHS that accounts for uncertainty in causal datasets especially when the confounders, treatments, mediators, and outcomes are not all observed in a single observational dataset. The main problem that is addressed here is quantifying the uncertainty in the estimation of causal effects when the treatment and the outcome are measured in two separate sources (potentially under different data-generating conditions) and the aim is to estimate the effect from a causal data fusion perspective. The paper introduces Bayesian Interventional Mean Processes for representing interventional distributions in the reproducing kernel Hilbert space. The method is applied to the task of Causal Bayesian Optimization where it is shown that the method converges faster as compared to other baseline methods. The main contribution of the paper is obtaining the uncertainty estimates for $E\[T\mid do(X) = x\]$.

**Limitations And Societal Impact:**

Given that the proposed approach aims to combine causal information from multiple datasets that may not all be collected under the same settings, it can have several potential impacts in fields like healthcare. The paper discusses this to some extent.

The limitations are discussed with respect to assuming complete knowledge of the causal graph. However, there could also be challenges with obtaining enough samples across the two sources, in which case there would be additional uncertainty because of representation bias. It would be helpful to elucidate the specific challenges and extensions in this case.

**Main Review:**

The idea of quantifying uncertainty about causal estimates especially when the treatment and the outcome are not observed in the same setting is of significant importance but relatively understudied. The proposed approach definitely builds in this direction. Relation to prior work is discussed and cited when the approach builds on ideas developed therein.

The claims are well-supported to the best of my knowledge.  The write-up is generally clear. While the example on estimating the effect of statin drug dosage on cancer volume is helpful, it would also help the reader to present some of the real-world settings where the approach may not succeed as expected, for example, in the case of recommender systems with high-dimensional confounding and higher uncertainty about the causal graph as well.

The results will aid practitioners in quantifying the uncertainties with respect to the causal effect estimates as well as guide in better data-collection practices if the uncertainties span over a larger region and estimating the effects with data from two different sources may not be accurate. It would help to address/analyze with the setting where the data is from a single source but the causal graph is uncertain, this can be an interesting future direction as well.




**Time Spent Reviewing:**

9

---

> ### Author Response · Authors · 2021-08-10
> **Response to reviewer wLHr**
>
> > “it would also help the reader to present some of the real-world settings where the approach may not succeed as expected, for example, in the case of recommender systems with high-dimensional confounding and higher uncertainty about the causal graph as well.”
>
>
> We thank the reviewer for pointing out this potential limitation in terms of uncertainty over the causal graph. However, our work (as do many in this field) mainly focuses on the case where we have full or partial knowledge of the causal graph, which is sufficient enough for modelling the do-calculus. (See general comment #3). We also discuss this limitation at the end of the discussion section (see line [356-360]), in which a future direction of our method would consider the case where the causal graph is not/partially known.
>
> In terms of the problem with high dimensional confounders, we will explicitly discuss the limitations in the revised version e.g. high dimensional confounding makes do-calculus more challenging. This high dimensional problem constitutes a new problem in itself, however, it is out of the scope of this paper and hence we leave it for future work.
>
> ------------------------------------------------------------------------------------------------------------------------
>
> > “It would help to address/analyze with the setting where the data is from a single source but the causal graph is uncertain, this can be an interesting future direction as well.”
>
> We thank the reviewer for this suggestion. We assume that when the reviewer says “the graph itself is uncertain”, you mean that the underlying structural causal model (SCM) behind the causal variables is uncertain? If yes, this could indeed be an interesting direction to pursue using our proposed methods in the paper. However, we note that the simplified case where data is from a single source with a certain causal graph has been tackled in [Aglietti et al 2020] through “sampling” and fitting GPs, which we have described in our paper (see line [266 - 272]). The problem that we are solving in this paper is mainly focused on how to consolidate unmatched data from separate data sources and hence we will leave this interesting direction with uncertain graphs for future work.
>
> ------------------------------------------------------------------------------------------------------------------------
>
> > “The limitations are discussed with respect to assuming complete knowledge of the causal graph. However, there could also be challenges with obtaining enough samples across the two sources, in which case there would be additional uncertainty because of representation bias. It would be helpful to elucidate the specific challenges and extensions in this case.”
>
> As the reviewer rightfully points out, there are challenges due to limited data in our problem setup. However, we would like to point the reviewer to lines 273-291 (Ablation study) and especially Appendix C1.2, where we answer exactly this question. We note that in the cases where we have less data, utilizing a GP actually allows us to add additional uncertainty as can be seen through the “extrapolation” effect in Fig.5 (c) and Appendix C1.2. We will explicitly add this comment in the revised version to make it clearer as well as potential extensions in this case.
>
>
>
> We thank the reviewer again for their time to review our work and hope that the above has clarified their concerns. We also believe as the reviewer stated that “quantifying uncertainty about causal estimates especially when the treatment and the outcome are not observed in the same setting is of significant importance but relatively understudied” and hope the reviewer could revise their score.
>
>
> References:
>
> Aglietti et al. 2020 Causal Bayesian Optimization, Proceedings of the Twenty Third International Conference on Artificial Intelligence and Statistics, PMLR 108:3155-3164, 2020.

---

> > ### Comment · Reviewer_wLHr · 2021-08-30
> > **Response to authors of Paper6134**
> >
> > I would like to thank the authors for their feedback and clarifications. The authors answer most of the concerns raised, adding the specific limitations with high-dimensional confounders would be valuable. I keep my score the same.

---

### Author Response · Authors · 2021-08-10
**General comments to all the reviewers**

We thank the reviewers for their time to review and leaving feedback on our paper. Before addressing the individual reviews, we would like to highlight particularly the 3 messages below:

1) The motivation for developing a Bayesian model for causal data fusion model with application to Bayesian optimisation.

- While it is true that frequentist methods can also produce uncertainty estimates by taking relative frequencies of multiple models fitted on subsampled data, such as Random Forest, it is known that probabilities constructed this way tend to be biased and are not necessarily well-calibrated. (Sec 4.1 [Ref.1]). Moreover, as [Ref.2] pointed out, in the case of Bayesian optimisation, methods like Random Forests are good at exploitation but don’t perform well for exploration, which is also a key reason why we consider developing  GP-based Bayesian models to tackle our causal data fusion setting and the application of the resulting uncertainty estimates to the causal Bayesian optimisation problem.

- Bayesian models allow incorporating the domain knowledge through the selection of priors (in our case, the choice of covariance functions in Gaussian processes). This may be relevant in a range of applications, including medical science.

2) A general problem setup and solution for combining causal graphs

- We would like to emphasize that our proposed method is a general framework for combining causal data structures from two separate datasets while still accounting for uncertainty in each graph. The key assumption we make is that all paths from the treatment variable X to the target variable T are mediated via a common variable (in our case Y). Hence we would like to explicitly state here, that even though our setup partially resembles the instrumental variable (IV) regression setting, i.e. the availability of two separate datasets (in this case a DAG and conditional relationship). Here we consider a problem setup, where we can have a confounder between the IV and the treatment, which is not allowed in standard IV regression. We go into more depth in the response to reviewer 3 (first argument) and will add this distinction in the revised version of our paper.

3) The assumption on the causal graph:

- We would like to stress that we do not need to have full knowledge of the whole causal graph but rather only the information on the nodes which are required for do-calculus. This is a standard and widely used assumption, also known as the strong ignorability assumption. Hence for this paper, we restrict ourselves to this scenario and leave the cases where we have unobserved confounders for future work.


Ref.1 [Hüllermeier, Eyke, and Willem Waegeman. "Aleatoric and epistemic uncertainty in machine learning: An introduction to concepts and methods." Machine Learning 110.3 (2021): 457-506.]

Ref.2 [Greenhill, Stewart, et al. "Bayesian optimization for adaptive experimental design: A review." IEEE access 8 (2020): 13937-13948.]

---

### Decision · Program_Chairs · 2021-09-27

**Decision:**

Accept (Poster)

**Comment:**

All four reviewers advocate acceptance. I also recommend accepting the paper for its contributions to the emerging field of Bayesian causal inference.